# One Batch Is Enough: A Unified Dataset Condensation Framework for General Time Series Analysis

**Wei Shao** [1] **Ziquan Fang** [1] **Zheqi Lu** [1] **Yongfeng Su** [1] **Yuzhu Wang** [2] **Yunjun Gao** [3]

## Abstract

Time-series analysis is critical in real-world applications, yet the explosion of time-series data imposes severe burdens on storage and computational resources. Recently, dataset condensation has emerged as a promising data-centric solution by synthesizing compact yet informative datasets to replace large-scale raw data. However, existing methods are largely vision-centric, failing to capture unique temporal properties of time series, or task-specific, tightly coupling the condensed data to a particular downstream objective. As a result, these approaches suffer from feature mismatch and fail to generalize across diverse time-series tasks. To bridge this gap, we propose UniTSC, the first unified dataset condensation framework for general time-series analysis. UniTSC employs a multi-view hybrid encoder to capture task-invariant representations across temporal, spectral, and topological perspectives. Building upon this representation, we design a tri-space alignment paradigm that jointly aligns optimization trajectories, power spectral densities, and multivariate dependency structures, enabling comprehensive information preservation under extreme compression. Extensive experiments show that UniTSC retains up to 97.9% of downstream performance using as little as 0.01% of the original training data, within our experimental settings tied to standard sequence lengths, revealing that a single batch-equivalent budget ($< 128$ samples) is sufficient to capture the essential dynamics of complex time-series data.

[1]School of Software, Zhejiang University, Ningbo, China [2]College of Computer and Information Engineering, Tianjin Normal University, Tianjin [3]College of Computer Science, Zhejiang University, Hangzhou, China. Correspondence to: Ziquan Fang <zqfang@zju.edu.cn>.

*Proceedings of the 43rd International Conference on Machine Learning*, Seoul, South Korea. PMLR 306, 2026. Copyright 2026 by the author(s).

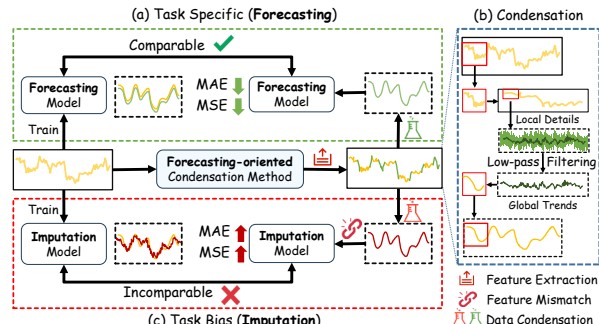

*Figure 1.* Task Bias in Time-Series Dataset Condensation

## 1. Introduction

Time-series analysis, encompassing tasks such as forecasting (Qiu et al., 2025), imputation (Yang et al., 2024), classification (Liu et al., 2025b), and anomaly detection (Zhang et al., 2022), underpins a wide range of real-world applications, from financial markets (Zhang et al., 2025) to network throughput (Dong et al., 2015). Meanwhile, the proliferation of IoT devices has led to an unprecedented explosion in data scale. Recent reports from the IoT Analytics project estimate over 21 billion connected devices by 2025 (IoT Analytics, 2025), generating massive volumes of continuous sensor data. However, data volume does not necessarily imply data value. According to the Veritas Global Databerg Report (Veritas Technologies, 2016), approximately 33% of stored data is redundant, obsolete, or trivial. Training models on such bloated, low-density datasets incurs severe burdens on storage, computation, and optimization (Strubell et al., 2019; Yu et al., 2024). To address these challenges, **dataset condensation** (also known as dataset distillation) has emerged as a promising data-centric paradigm (Lee et al., 2022; Cui et al., 2023). By synthesizing a compact yet informative dataset to replace the original one, condensation aims to reduce training cost while preserving the critical information required for downstream performance.

Early condensation techniques primarily relied on coreset selection, extracting representative subsets based on heuristic rules (Lucic et al., 2017; Chai et al., 2023) or geometric distance metrics (Har-Peled & Kushal, 2005; Feldman & Langberg, 2011). While intuitive, such selection-based approaches offer no optimality guarantees and often discard samples that are crucial for downstream learning (Lee &

Chung, 2024). More recently, optimization-based condensation methods, such as gradient matching (Zhao et al., 2021; Cazenavette et al., 2022) and kernel-based optimization (Zhou et al., 2022; Loo et al., 2023), have demonstrated remarkable success in synthesizing highly compact datasets. Despite this progress, *the majority of existing condensation research remains vision-centric*. Unlike images, which are static and spatially structured, time series exhibit complex temporal dependencies, non-stationarity, and periodic dynamics (Wu et al., 2025). As a result, directly transplanting image-based condensation techniques often fails to capture the intrinsic generative mechanisms of temporal data.

Several recent works, including CondTSF (Ding et al., 2024), CondTSC (Liu et al., 2024b), and TimeDC (Miao et al., 2024), have begun to explore dataset condensation for time series. However, *these methods remain fundamentally constrained by **task-specific bias***, as the condensation process is intrinsically coupled with a particular downstream objective. As illustrated in Fig. 1(b), forecasting-oriented approaches (e.g., CondTSF) minimize prediction error, effectively acting as an implicit low-pass filter (see **Appendix D.6** for qualitative evidence). While such bias preserves global trends sufficient for forecasting in Fig. 1(a), it aggressively suppresses high-frequency local variations. This selective information retention leads to the problem shown in Fig. 1(c): when applied to imputation tasks that rely heavily on local volatility, the absence of fine-grained details results in significant performance degradation. More broadly, this intrinsic task bias induces unavoidable feature mismatch, preventing existing paradigms from serving as a unified solution across diverse time-series analysis tasks.

To overcome these limitations, we advocate a paradigm shift from **task-fitting** to **mechanism-preserving** condensation. While downstream objectives vary, the physical generation mechanism of time series remains invariant. Specifically, such mechanisms are governed by the interaction of three orthogonal factors: temporal evolution (non-stationary dynamics), spectral periodicity (recurring patterns), and multivariate coupling (inter-variable dependencies). Motivated by this observation, we propose **UniTSC**, a unified dataset condensation framework that optimizes synthetic data through a **tri-space alignment paradigm**. Specifically, UniTSC employs a multi-view hybrid encoder to extract task-invariant representations from temporal, spectral, and topological perspectives, forming a physically complete shared feature space. Building upon this representation, we further design a unified alignment objective that matches synthetic and real data across three complementary spaces: (i) parameter space, to reconstruct learning trajectories driven by temporal dynamics; (ii) frequency space, to preserve governing periodic laws; and (iii) topology space, to retain the latent graph structure of multivariate dependencies. By preserving the underlying generative mechanisms, UniTSC naturally generalizes across diverse time-series analysis tasks, effectively mitigating the feature mismatch induced by task bias.

Overall, our contributions are summarized as follows:

- To the best of our knowledge, we propose the first unified dataset condensation framework for general time-series analysis, breaking the limitations of task-specific designs and alleviating feature mismatch caused by task bias.

- To extract task-shared representations, we design a multi-view encoder involving temporal, spectral, and topological domains via multi-scale patching, adaptive Fourier modeling, and variable-level topology learning.

- We introduce a tri-space alignment paradigm that matches synthetic and real data in parameter, frequency, and topology spaces, enabling comprehensive information preservation under extreme condensation.

- Extensive experiments across four mainstream time-series tasks on 22 datasets and nine representative baselines demonstrate that UniTSC consistently achieves state-of-the-art performance retention, while exhibiting strong cross-architecture generalization, superior data efficiency, and robust performance across different data scales.

## 2. Related Work

Due to space limitations, comprehensive reviews of **general time-series analysis** and **standard dataset condensation** are deferred to **Appendix A**. In this section, we focus on the most related topic of **time-series dataset condensation**.

Compared with the mature literature in the image domain, dataset condensation for time series remains at an early stage, with most existing methods exhibiting **task-specific bias**. Specifically, CondTSF (Ding et al., 2024) is designed for forecasting and optimizes regression objectives through gradient and value matching. Such optimization implicitly acts as a low-pass filter, prioritizing global trends while suppressing high-frequency variations. While effective for long-horizon forecasting, this bias limits its ability to preserve local continuity that is crucial for reconstruction tasks such as imputation. Similarly, CondTSC (Liu et al., 2024b) targets classification by performing dual-domain matching to enhance inter-class separability. However, this objective inevitably reshapes the data distribution, retaining only discriminative features while discarding non-discriminative background patterns. As a result, this method fails to model normality required by anomaly detection tasks. Although TimeDC (Miao et al., 2024) attempts to unify forecasting and classification via curriculum-based trajectory matching, its condensation process still relies on task-specific supervision signals to guide optimization. Moreover, TimeDC adopts a fixed trend-seasonality decomposition that either treats variables independently or captures only simple pairwise correlations, thereby overlooking the multivariate topological couplings inherent in real-world time series.

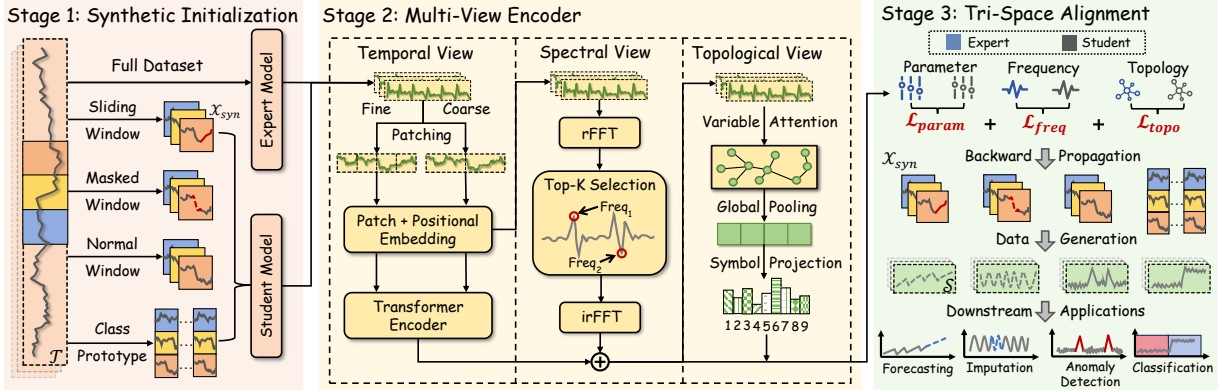

*Figure 2.* The overall framework of UniTSC. The method proceeds in three stages: **(i) Task-Adaptive Synthetic Initialization:** We initialize latent parameters $\mathcal{X}_{syn}$ by utilizing task-adaptive priors. **(ii) Multi-View Hybrid Encoder:** A shared backbone extracts holistic representations by integrating temporal, spectral, and topological views. **(iii) Tri-Space Alignment:** We optimize $\mathcal{X}_{syn}$ by minimizing the divergence between student and expert models across parameter, frequency, and topology spaces, yielding the condensed dataset $\mathcal{S}$.

These methods share a common limitation: *the condensation objective is tightly coupled with downstream tasks, preventing comprehensive preservation of the underlying generative mechanisms.* This motivates a unified framework that aligns time-series data across temporal, spectral and topological dimensions within a single methodology.

## 3. Preliminaries

The frequently used notations are shown in **Appendix B.1**.

**Definition 1 (Time Series).** A time series is an ordered sequence of observations indexed by time, denoted as $\mathbf{X} = (\mathbf{x}_1, \mathbf{x}_2, \ldots, \mathbf{x}_L)$, where $L$ is the sequence length. Each observation $\mathbf{x}_i \in \mathbb{R}^C$ is a $C$-dimensional measurement.

Following standard conventions in general time-series analysis (Wu et al., 2023; Zhou et al., 2023), we consider four representative downstream tasks: **forecasting, imputation, classification**, and **anomaly detection**, which collectively capture the core challenges of temporal modeling. Accordingly, we define a time-series dataset as follows.

**Definition 2 (Time Series Dataset).** Given a downstream task, a time-series dataset is denoted by $\mathcal{T} = \{(\mathbf{X}_i, \mathbf{Y}_i)\}_{i=1}^{N}$, where $N$ is the number of samples, $\mathbf{X}_i \in \mathbb{R}^{L \times C}$ represents an input time series segment, and $\mathbf{Y}_i$ denotes the corresponding task-specific supervision signal.

To unify distinct tasks under this formulation, we adapt $\mathbf{Y}_i$ as follows: for forecasting, $\mathbf{Y}_i \in \mathbb{R}^{H \times C}$ corresponds to the future horizon; for classification, $\mathbf{Y}_i \in \{0, \ldots, K-1\}$ denotes the class label; for imputation and anomaly detection, $\mathbf{Y}_i \in \mathbb{R}^{L \times C}$ aligns with $\mathbf{X}_i$ as a reconstruction objective.

**Definition 3 (Time Series Dataset Condensation).** Given a large-scale dataset $\mathcal{T} = \{(\mathbf{X}_i, \mathbf{Y}_i)\}_{i=1}^{N}$, time-series dataset condensation aims to synthesize a significantly smaller yet informative dataset $\mathcal{S} = \{(\tilde{\mathbf{X}}_j, \tilde{\mathbf{Y}}_j)\}_{j=1}^{M}$, where $M \ll N$. Let $\phi_\theta$ denote a neural network parameterized by $\theta$. The core objective is to learn an optimal synthetic set $\mathcal{S}$ such that

a model trained on $\mathcal{S}$ achieves generalization performance comparable to one trained on $\mathcal{T}$. This is formally cast as the following bi-level optimization problem:

$$\min_{\mathcal{S}} \mathcal{L}_{\text{val}}(\mathcal{T}_{\text{val}}; \theta_{\mathcal{S}}^*) \quad \text{s.t.} \quad \theta_{\mathcal{S}}^* = \arg\min_{\theta} \mathcal{L}_{\text{train}}(\mathcal{S}; \theta), \quad (1)$$

where $\theta_{\mathcal{S}}^*$ represents the optimal parameters trained on $\mathcal{S}$, and $\mathcal{L}_{\text{train}}$, $\mathcal{L}_{\text{val}}$ denote the training and evaluation losses.

## 4. Methodology

As illustrated in Fig. 2, UniTSC consists of three tightly coupled components: (i) Task-Adaptive Synthetic Initialization, (ii) a Multi-View Hybrid Encoder, and (iii) a Tri-Space Alignment Paradigm. Together, these components enable UniTSC to preserve the underlying generative mechanisms of time series while avoiding the task bias.

### 4.1. Task-Adaptive Synthetic Initialization

To accommodate the heterogeneity of time-series tasks, we introduce a task-adaptive synthetic initialization strategy with a weak task prior. Specifically, it acts merely as a structural adapter to respect distinct data formats, ensuring the synthetic data is physically compatible with downstream tasks without altering the unified condensation mechanism.

**Continuous initialization for sequence modeling tasks.** For forecasting, imputation, and anomaly detection, time-series samples exhibit strong temporal dependencies and continuity. Modeling synthetic data as independent and discrete instances—as is common in image-based condensation—violates inductive bias and disrupts temporal coherence. To preserve temporal continuity, we parameterize the synthetic data as a globally continuous latent tensor $\mathcal{X}_{syn} \in \mathbb{R}^{L_{syn} \times C}$. The latent length $L_{syn}$ is governed by $L_{syn} = W + \tau(M - 1)$, where $W$, $M$, and $\tau$ denote the window size, sample budget, and sliding stride, respectively. As illustrated in Fig. 2, we first initialize $\mathcal{X}_{syn}$ by applying a sliding window over the real dataset to incorporate

task-adaptive priors.

During the subsequent optimization loop, the final synthetic dataset $\mathcal{S}$ is dynamically constructed by applying a sliding-window operator over $\mathcal{X}_{syn}$, yielding $M$ overlapping segments. This design implicitly regularizes the condensation process by enforcing temporal coupling across synthetic samples, encouraging smooth and evolving dynamics rather than isolated, instance-specific patterns.

**Discrete prototype initialization for classification.** In contrast, classification tasks typically assume sample-wise independence and rely on explicit class supervision. Accordingly, we adopt a class-balanced discrete parameterization. Specifically, we initialize the synthetic dataset as a set of learnable class prototypes $\mathcal{X}_{syn} \in \mathbb{R}^{(K \times spc) \times L \times C}$, together with fixed labels $\mathcal{Y}_{syn}$, where $K$ is the number of classes, $spc$ is the number of synthetic samples per class. The label set $\mathcal{Y}_{syn}$ is constructed as a balanced class-repeated sequence, ensuring uniform class coverage and providing stable supervision throughout the condensation process.

### 4.2. Multi-View Hybrid Encoder

**Design Motivation.** To enable a unified dataset condensation framework, we first clarify the fundamental components that constitute a general-purpose time-series representation. From a generative standpoint, multivariate time series are governed by a small set of complementary latent factors, a view widely supported by classical signal processing and system modeling theories (Brockwell & Davis, 2009). In particular, these factors can be characterized along three largely orthogonal dimensions: (1) *local temporal dynamics*, such as trends and short-term volatility, which are predominantly shaped by auto-regressive mechanisms; (2) *global spectral properties*, including periodicity and energy distribution, which are naturally described in the frequency domain; and (3) *spatial geometric structures* (e.g., variable correlations) determined by the underlying system topology (Wu et al., 2020). Preserving any single factor in isolation is insufficient for robust downstream generalization. Instead, a representation that jointly captures temporal evolution, spectral regularities, and multivariate couplings is essential for supporting diverse time-series analysis tasks.

**(1) Multi-Scale Temporal View.** A fundamental challenge in time-series dataset condensation lies in compressing high-entropy temporal dynamics into a limited synthetic budget. Relying on a single temporal resolution introduces an information bottleneck: fine-grained volatility may be suppressed, or long-term trends may be inadequately captured. To alleviate this issue, we adopt a *multi-scale patching mechanism* that preserves temporal information across multiple resolutions during representation learning.

Formally, given the instance-normalized input series $\mathbf{X} \in \mathbb{R}^{L \times C}$, where $L$ denotes the sequence length and $C$ denotes

the number of channels, we unfold the series into two distinct granularities, $\mathbf{X}^{(1)}$ and $\mathbf{X}^{(2)}$. These multi-scale patch sequences are projected into the latent space to yield embeddings $\mathbf{E}^{(k)} \in \mathbb{R}^{C \times P_k \times D}$, where $P_k$ is the number of patches and $D$ is the embedding dimension. Subsequently, these embeddings are processed by a weight-shared Transformer encoder augmented with positional embedding $\mathbf{W}_{pos}^{(k)}$:

$$\mathbf{H}^{(k)} = \text{Encoder}(\mathbf{E}^{(k)} + \mathbf{W}_{pos}^{(k)}), \ k \in \{1, 2\}. \quad (2)$$

Within the condensation framework, this multi-scale design enforces a form of *scale-invariant inductive bias*, enabling the synthetic data to jointly encode coarse-grained global contexts and fine-grained local dynamics. As a result, the condensed dataset achieves improved robustness and generalization across downstream tasks that rely on temporal patterns at different resolutions.

**(2) Adaptive Spectral View.** While Parseval's theorem (Oppenheim, 1999) ensures that the total energy is preserved between the time and frequency domains, standard Transformers often exhibit spectral bias, tending to capture low-frequency trends while overlooking high-frequency details or global periodic structures (Rahaman et al., 2019). To address this limitation, we introduce a *Fourier Mix Block* after the patch embedding layer, explicitly enforcing consistency between synthetic and real data in the frequency domain.

Concretely, given patch embeddings $\mathbf{E}$, we apply a Real Fast Fourier Transform (rFFT) along the patch dimension to obtain the spectrum $\mathcal{F} \in \mathbb{C}^{C \times F \times D}$, where $F = \lfloor P/2 \rfloor + 1$ denotes the number of valid frequency components. To capture dominant periodic patterns while preserving high-frequency details, we select the top-$k$ components separately from both low-frequency and high-frequency bands to preserve global trends and local variations. Let $\mathcal{K}$ denote the selected indices, the spectral modulation is applied to these indices using a learnable complex tensor $\mathbf{W}_{freq} \in \mathbb{C}^{F \times D}$:

$$\tilde{\mathcal{F}}_i = \begin{cases} \mathcal{F}_i \odot \mathbf{W}_{freq,i} & \text{if } i \in \mathcal{K}, \\ 0 & \text{otherwise}, \end{cases} \quad (3)$$

where $\odot$ denotes element-wise complex multiplication. The modulated spectrum is then transformed back to the time domain via inverse rFFT and fused through a residual connection: $\mathbf{H}' = \mathbf{H} + \text{Dropout}(\text{irFFT}(\tilde{\mathcal{F}}))$. Note that to align with the multi-scale design, this spectral modulation and residual connection are applied independently to both scales $\mathbf{H}^{(1)}$ and $\mathbf{H}^{(2)}$, which are subsequently concatenated to form the unified temporal embedding $\mathbf{Z}_{time}$.

This block effectively functions as a learnable adaptive band-pass filter, selectively amplifying critical periodic signals. In the context of data condensation, it imposes a strong inductive bias that preserves the authentic power spectral density of synthetic data, mitigating the common loss of high-frequency fidelity in condensed datasets.

**(3) Topological Relational View.** While channel-independent (CI) architectures excel in forecasting (Chen et al., 2024), they impose limitations for dataset condensation. By treating variables in isolation, CI-based methods explicitly discard cross-variable dependencies that characterize the underlying physical system. Synthesizing data without these constraints results in disjoint datasets that fail to capture authentic multivariate coupling structures. To mitigate this, we introduce a *Topological Relation Branch*, which shifts the alignment objective from unstable numerical matching to robust relational regime alignment.

Specifically, we first aggregate the temporal embeddings $\mathbf{Z}_{time}$ along the patch dimension to obtain variable-level summaries, and reconstruct the underlying implicit graph structure via a self-attention mechanism. The relational features across variables are then pooled to produce a global state vector $\mathbf{s}$, which is projected into a discrete topology space to yield modal distribution logits:

$$Q = \mathbf{s}\mathbf{W}_{topo} + \mathbf{b}_{topo}, \tag{4}$$

where $\mathbf{W}_{topo}$ and $\mathbf{b}_{topo}$ are learnable projection parameters.

Within the condensation framework, this topological projection acts as a crucial information bottleneck: it encourages the synthetic dataset to ignore stochastic fluctuations and instead match discrete, global interaction regimes. By aligning these topological distributions, the multivariate coupling structures encoded in the attention-induced graph are explicitly preserved. As a result, the structure-preserving reconstruction ensures that the synthesized data retains robust geometric properties, satisfying diverse feature-level inductive biases required by downstream tasks.

### 4.3. Tri-Space Alignment Paradigm

**Design Motivation.** Motivated by the generative perspective established in Sec. 4.2, we introduce the Tri-Space Alignment Paradigm, a principled condensation objective that enforces consistency between real and synthetic data across three complementary and largely orthogonal spaces. Specifically, we impose alignment constraints in: (1) *parameter space*, to capture learning dynamics and prevent long-horizon trajectory drift; (2) *frequency space*, to preserve global spectral laws and mitigate spectral bias where high-frequency details are absorbed by low-frequency overfitting; and (3) *topology space*, to retain multivariate coupling regimes and prevent the loss of underlying physical structure in synthetic samples. By jointly aligning these three spaces, UniTSC moves beyond local gradient matching and instead approximates the underlying generative mechanisms that govern time-series data, enabling stable optimization and robust generalization under extreme data compression.

**(1) Parameter Space.** The generalization capability of time series analysis models is heavily dependent on the curvature and stability of their optimization path. To capture these long-term learning dynamics, we adopt a trajectory-based matching paradigm.

Let $\boldsymbol{\theta}_t^* \rightarrow \boldsymbol{\theta}_{t+T}^*$ denote a parameter trajectory sampled from the expert buffer $\mathcal{B}$. We initialize the student model state $\hat{\boldsymbol{\theta}}_0$ directly from the expert's starting point $\boldsymbol{\theta}_t^*$. Thus, the update process of the student model on the synthetic data $\mathcal{S}$ over $T$ steps of gradient descent can be viewed as an accumulation operator. Our objective is to minimize the normalized Euclidean distance between the student's parameters $\hat{\boldsymbol{\theta}}_T$ and the expert's target parameters $\boldsymbol{\theta}_{t+T}^*$:

$$\mathcal{L}_{param} = \frac{||\hat{\boldsymbol{\theta}}_T - \boldsymbol{\theta}_{t+T}^*||_2^2}{||\boldsymbol{\theta}_t^* - \boldsymbol{\theta}_{t+T}^*||_2^2}. \tag{5}$$

Minimizing $\mathcal{L}_{param}$ essentially enforces the synthetic data to internalize the long-range optimization dynamics of the expert. Unlike greedy single-step matching, this trajectory-based alignment mitigates the error accumulation effect during the training process. It ensures that the student model converges to the same optimal solution space as the expert, thereby guaranteeing that the learned representations are physically consistent with the underlying generative mechanism of the real time series.

**(2) Frequency Space.** Parameter matching inherently minimizes temporal-view errors, leading to spectral oversmoothing, sacrificing high-frequency textures for low-frequency trends. To bridge this gap, we impose a frequency-space constraint as a physics-informed global regularizer, explicitly enforcing preservation of authentic power spectral density.

Let $\mathbf{Y}_S$ and $\mathbf{Y}_E$ denote the outputs of the student and expert models. We first apply instance normalization to decouple spectral shape from scale intensity, followed by an rFFT to obtain the complex spectral coefficients of the valid frequency components, denoted as $\mathbf{Z}_S = \text{rFFT}(\text{Norm}(\mathbf{Y}_S))$ and $\mathbf{Z}_E = \text{rFFT}(\text{Norm}(\mathbf{Y}_E))$. Instead of optimizing amplitude and phase separately, which risks numerical instability due to phase wrapping, we enforce strict alignment in the Cartesian coordinate system. We minimize the divergence between the real ($\mathfrak{R}$) and imaginary ($\mathfrak{I}$) components:

$$\mathcal{L}_{freq} = ||\mathfrak{R}(\mathbf{Z}_S) - \mathfrak{R}(\mathbf{Z}_E)||_F^2 + ||\mathfrak{I}(\mathbf{Z}_S) - \mathfrak{I}(\mathbf{Z}_E)||_F^2. \tag{6}$$

Notably, for classification tasks where $\mathbf{Y}$ represents class probabilities, this objective is omitted. In other scenarios, this physical constraint acts as a global regularizer, compelling the student to capture the exact energy distribution and temporal positioning of the expert, thereby preventing the loss of frequency information during condensation.

**(3) Topology Space.** While parameter and frequency alignments capture temporal dynamics, they often neglect inter-variable dependencies. In dataset condensation, this omission is catastrophic: synthetic samples degenerate into disjoint features, losing the structural coherence of real manifold. To prevent this, we enforce alignment in the topology space, explicitly preserving the manifold structure.

*Table 1.* Summary of Experiment Benchmarks (see Table 6 in **Appendix C.1** for details)

| Tasks | Benchmarks | Metrics | Series Length |
|---|---|---|---|
| Forecasting | ETT (4 subsets), Weather, Electricity, Traffic | MSE, MAE | 96 |
| Imputation | ETT (4 subsets), Weather, Electricity | MSE, MAE | 96 |
| Classification | UEA (10 subsets) | Accuracy | 29∼1751 |
| Anomaly Detection | SMAP, PSM, SMD, SWaT, MSL | Precision, Recall, F1-Score | 100 |

Leveraging the topological relational view, we utilize the extracted discrete regime distribution $\mathbf{Q}$ as a structural fingerprint to characterize the topological state. Specifically, we employ KL divergence to directly transfer the expert's implicit structural knowledge to the student:

$$\mathcal{L}_{topo} = \text{KL}\left(\sigma(\mathbf{Q}_E) \,\|\, \sigma(\mathbf{Q}_S)\right), \tag{7}$$

where $\mathbf{Q}_E$ and $\mathbf{Q}_S$ represent the topological states of the expert and student, and $\sigma(\cdot)$ denotes the softmax function normalizing the regime logits into a probability distribution. Minimizing $\mathcal{L}_{topo}$ is equivalent to structural regularization: it forces the student model to infer a variable interaction graph consistent with the real data, thereby preventing overfitting to spurious correlations caused by sample sparsity.

To theoretically justify the design of our spectral and topological alignment modules, we formalize the limitations of naive parameter space alignment.

**Proposition 4.1** (**Insufficiency of Parameter Space Alignment**). *Assuming the gradient flow follows the Neural Tangent Kernel ([Rahaman et al., 2019](#)) dynamics, the parameter alignment objective fails to reduce discrepancies in high-frequency and topological domains. Consequently, the distributional discrepancy $\mathcal{D}(\mathcal{S}, \mathcal{T})$ is lower-bounded by the irreducible errors in these domains:*

$$\mathcal{D}(\mathcal{S}, \mathcal{T}) \geq C_1 \cdot \mathcal{E}_{spec}(\mathcal{S}, \mathcal{T}) + C_2 \cdot \mathcal{E}_{topo}(\mathcal{S}, \mathcal{T}), \tag{8}$$

*where $C_1, C_2 > 0$ are constants related to the curvature of the loss landscape, $\mathcal{E}_{spec} = \|\mathcal{F}(\mathcal{S})_{high} - \mathcal{F}(\mathcal{T})_{high}\|^2$ denotes the divergence in high-frequency spectral components, and $\mathcal{E}_{topo}(\mathcal{S}, \mathcal{T}) = \|\Sigma_{\mathcal{S}} - \Sigma_{\mathcal{T}}\|_F^2$ quantifies the mismatch in second-order dependency structures.*

Proposition 4.1 theoretically explains the experimental observations in ablation study (**Appendix D.1**). It implies that without explicit regularization in frequency and topology domains, parameter alignment alone is insufficient to capture high-frequency details and complex multivariate dependencies. The detailed proof is provided in **Appendix B.2**.

### 4.4. Optimization Objective

To ensure holistic consistency across the defined manifolds, we integrate the constraints from the parameter, frequency, and topology spaces into a tri-space alignment paradigm. Specifically, the dataset condensation process is formalized as seeking the optimal synthetic set $\mathcal{S}$ that minimizes the expected divergence over expert trajectories sampled from buffer $\mathcal{B}$. The total objective function is defined as:

$$\min_{\mathcal{S}} \mathbb{E}_{(\boldsymbol{\theta}_t, \boldsymbol{\theta}_{t+T}) \sim \mathcal{B}} \left[ w_1 \mathcal{L}_{param} + w_2 \mathcal{L}_{freq} + w_3 \mathcal{L}_{topo} \right]. \tag{9}$$

Here, $w_i$ denotes the weight balancing the contribution of each alignment constraint in the tri-space objective. The complete optimization procedure of UniTSC is summarized in Alg. 1 (see **Appendix B.3** for details).

To theoretically validate the paradigm, we bound the generalization error of a student model trained on $\mathcal{S}$ as follows.

**Theorem 4.2** (**Generalization Error Bound**). *Let $\mathcal{R}(\theta)$ denote the expected risk on the real data distribution. Assume the loss function is $G$-Lipschitz and $\beta$-smooth. For a student model $\hat{\theta}_T$ updated for $T$ steps on the synthetic set $\mathcal{S}$ and an expert model $\theta_T^*$ trained on the real set $\mathcal{T}$, the generalization gap is bounded by:*

$$\mathcal{R}(\hat{\theta}_T) - \mathcal{R}(\theta_T^*) \leq G\eta \sum_{t=0}^{T-1} (1 + \eta\beta)^{T-1-t} \mathcal{E}_{align}^{(t)}, \tag{10}$$

*where $G$ is the Lipschitz constant, $\eta$ is the learning rate, and $\mathcal{E}_{align}^{(t)}$ represents the gradient estimation error at step $t$, which is assumed to be bounded by the weighted tri-space alignment loss: $\lambda_1 \sqrt{\mathcal{L}_{param}} + \lambda_2 \sqrt{\mathcal{L}_{freq}} + \lambda_3 \sqrt{\mathcal{L}_{topo}}$.*

Theorem 4.2 demonstrates that generalization capability is upper-bounded by the tri-space alignment. Crucially, minimizing the auxiliary constraints $\mathcal{L}_{freq}$ and $\mathcal{L}_{topo}$ tightens the bound on gradient variance. Physically, this confines optimization trajectory to the valid manifold, preventing divergence into spurious loss valleys and minimizing the generalization gap. The detailed proof is provided in **Appendix B.4**. We note that the assumptions of NTK dynamics and Lipschitz smoothness are standard theoretical tools ([Nguyen et al., 2021b](#); [Tan & Slade, 2025](#)) used here to formally motivate our tri-space design by showing how spectral and topological alignment directly tighten the generalization error bound, rather than providing a tight bound under all realistic conditions.

## 5. Experiments

### 5.1. Experimental Settings

**Datasets.** We conduct comprehensive evaluations on 22 real-world datasets spanning four distinct downstream tasks. The dataset statistics are summarized in Table 1, with detailed descriptions provided in **Appendix C.1**.

**Baselines.** We compare UniTSC with nine baseline methods categorized into three groups: (i) *coreset selection strategies*, including Random Selection, K-Center ([Farahani &](#)

*Table 2.* Average Performance Comparison in Forecasting (**red**: best, blue: second). See Table 10 in **Appendix D.7** for full results.

| Models | Random | | K-Center | | Herding | | DC | | MTT | | DM | | CondTSF | | TimeDC | | **UniTSC** | | *Full* | | *TimesNet* | | *GPT4TS* | |
| Metric | MAE | MSE | MAE | MSE | MAE | MSE | MAE | MSE | MAE | MSE | MAE | MSE | MAE | MSE | MAE | MSE | MAE | MSE | MAE | MSE | MAE | MSE | MAE | MSE |
|---|---|---|---|---|---|---|---|---|---|---|---|---|---|---|---|---|---|---|---|---|---|---|---|---|
| ETTh1 | 0.614 | 0.774 | 0.613 | 0.765 | 0.616 | 0.767 | 0.549 | 0.644 | 0.534 | 0.616 | 0.588 | 0.681 | 0.486 | 0.519 | 0.494 | 0.534 | **0.442** | **0.475** | *0.437* | *0.452* | *0.433* | *0.437* | *0.416* | *0.411* |
| ETTh2 | 0.576 | 0.584 | 0.538 | 0.517 | 0.534 | 0.515 | 0.479 | 0.498 | 0.474 | 0.490 | 0.496 | 0.514 | 0.435 | 0.408 | 0.419 | 0.403 | **0.397** | **0.380** | *0.392* | *0.369* | *0.413* | *0.398* | *0.379* | *0.337* |
| ETTm1 | 0.585 | 0.798 | 0.582 | 0.789 | 0.588 | 0.808 | 0.517 | 0.572 | 0.511 | 0.559 | 0.511 | 0.569 | 0.467 | 0.444 | 0.512 | 0.463 | **0.406** | **0.411** | *0.388* | *0.381* | *0.391* | *0.374* | *0.371* | *0.330* |
| ETTm2 | 0.510 | 0.477 | 0.451 | 0.418 | 0.448 | 0.414 | 0.357 | 0.317 | 0.373 | 0.333 | 0.366 | 0.326 | 0.347 | 0.295 | 0.355 | 0.316 | **0.317** | **0.258** | *0.306* | *0.243* | *0.309* | *0.252* | *0.301* | *0.229* |
| Weather | 0.407 | 0.381 | 0.405 | 0.378 | 0.406 | 0.378 | 0.370 | 0.377 | 0.351 | 0.358 | 0.346 | 0.334 | 0.334 | 0.306 | 0.297 | 0.279 | **0.278** | **0.255** | *0.268* | *0.242* | *0.262* | *0.224* | *0.249* | *0.207* |
| Electricity | 0.458 | 0.355 | 0.457 | 0.358 | 0.461 | 0.356 | 0.435 | 0.336 | 0.412 | 0.314 | 0.424 | 0.325 | 0.366 | 0.274 | 0.367 | 0.277 | **0.348** | **0.264** | *0.304* | *0.216* | *0.287* | *0.183* | *0.252* | *0.154* |
| Traffic | 0.539 | 0.820 | 0.541 | 0.825 | 0.541 | 0.829 | 0.535 | 0.818 | 0.517 | 0.771 | 0.528 | 0.793 | 0.511 | 0.770 | 0.523 | 0.794 | **0.470** | **0.737** | *0.382* | *0.615* | *0.331* | *0.613* | *0.289* | *0.402* |

*Table 3.* Average Performance Comparison in Imputation (**red**: best, blue: second). See Table 11 in **Appendix D.7** for full results.

| Models | Random | | K-Center | | Herding | | DC | | MTT | | DM | | CondTSF | | TimeDC | | **UniTSC** | | *Full* | | *TimesNet* | | *GPT4TS* | |
| Metric | MAE | MSE | MAE | MSE | MAE | MSE | MAE | MSE | MAE | MSE | MAE | MSE | MAE | MSE | MAE | MSE | MAE | MSE | MAE | MSE | MAE | MSE | MAE | MSE |
|---|---|---|---|---|---|---|---|---|---|---|---|---|---|---|---|---|---|---|---|---|---|---|---|---|
| ETTh1 | 0.457 | 0.474 | 0.451 | 0.476 | 0.427 | 0.419 | 0.365 | 0.300 | 0.336 | 0.261 | 0.392 | 0.335 | 0.313 | 0.236 | 0.326 | 0.244 | **0.295** | **0.205** | *0.246* | *0.140* | *0.187* | *0.078* | *0.173* | *0.069* |
| ETTh2 | 0.331 | 0.217 | 0.295 | 0.194 | 0.292 | 0.186 | 0.273 | 0.166 | 0.264 | 0.158 | 0.285 | 0.175 | 0.244 | 0.139 | 0.257 | 0.149 | **0.185** | **0.082** | *0.168* | *0.068* | *0.146* | *0.050* | *0.141* | *0.048* |
| ETTm1 | 0.343 | 0.292 | 0.342 | 0.291 | 0.344 | 0.294 | 0.298 | 0.217 | 0.299 | 0.218 | 0.304 | 0.225 | 0.279 | 0.183 | 0.285 | 0.190 | **0.226** | **0.128** | *0.160* | *0.062* | *0.109* | *0.028* | *0.105* | *0.027* |
| ETTm2 | 0.265 | 0.141 | 0.227 | 0.116 | 0.226 | 0.115 | 0.184 | 0.084 | 0.178 | 0.080 | 0.196 | 0.091 | 0.155 | 0.070 | 0.160 | 0.074 | **0.130** | **0.042** | *0.109* | *0.032* | *0.089* | *0.022* | *0.085* | *0.021* |
| Weather | 0.196 | 0.141 | 0.191 | 0.141 | 0.182 | 0.127 | 0.110 | 0.074 | 0.110 | 0.072 | 0.130 | 0.087 | 0.096 | 0.065 | 0.087 | 0.059 | **0.069** | **0.038** | *0.057* | *0.033* | *0.054* | *0.030* | *0.057* | *0.031* |
| Electricity | 0.313 | 0.182 | 0.312 | 0.182 | 0.313 | 0.183 | 0.285 | 0.155 | 0.298 | 0.161 | 0.287 | 0.156 | 0.276 | 0.149 | 0.281 | 0.153 | **0.270** | **0.140** | *0.245* | *0.116* | *0.211* | *0.092* | *0.207* | *0.091* |

Hekmatfar, 2009), and Herding (Welling, 2009); (ii) *general dataset condensation methods*, namely DC (Zhao et al., 2021), MTT (Cazenavette et al., 2022), and DM (Zhao & Bilen, 2023); and (iii) *time series dataset condensation methods*, including CondTSF (Ding et al., 2024), CondTSC (Liu et al., 2024b), and TimeDC (Miao et al., 2024). Additionally, to establish performance upper bound, we report the results of two state-of-the-art models, TimesNet (Wu et al., 2023) and GPT4TS (Zhou et al., 2023), trained on the full datasets. Detailed descriptions are available in **Appendix C.2**.

**Implementations.** More details about metrics and implementations can refer to **Appendix C.3** and **Appendix C.4**.

### 5.2. Forecasting Task

**Setups.** Fixing the condensed budget $M$ to 96 samples (matching the look-back window), we evaluate forecasting performance across three horizons $\{96, 192, 336\}$.

**Results.** As reported in Table 2, UniTSC consistently outperforms all baselines, achieving an average MSE reduction of 7% relative to the second-best method. Remarkably, despite relying on only 96 condensed samples—approximately 0.6% of the original training set—our method preserves, on average, 92.4% of the performance of full-data training, and in the best case, up to 98.8%. These results support our core hypothesis: for general time series analysis, a single batch of high-quality synthetic data is sufficient to capture the essential properties of the underlying distribution. Consequently, UniTSC achieves performance comparable to full-data training while reducing data dependency.

### 5.3. Imputation Task

**Setups.** Fixing the condensed budget $M$ to 96 (matching the input window), we evaluate imputation performance across four masking ratios $\{12.5\%, 25\%, 37.5\%, 50\%\}$.

**Results.** As summarized in Table 3, UniTSC consistently achieves state-of-the-art performance across all datasets, retaining up to 90.6% of the performance obtained with the full dataset. Compared to the forecasting task, performance degradation is more noticeable, reflecting the inherently fine-grained nature of imputation: accurate reconstruction requires rich local details and high data diversity. Capturing the complexity of random masking patterns using only a limited set of condensed samples poses an intrinsic challenge. Nevertheless, UniTSC consistently outperforms all baselines, demonstrating that its multi-view encoder and tri-space alignment mechanisms effectively preserve critical reconstruction information. This enables near-optimal recovery even under a limited sample budget, validating the efficacy of our approach for fine-grained imputation tasks.

### 5.4. Classification Task

**Setups.** We employ a 5-shot condensation setting (i.e., 5 samples per class) across 10 diverse datasets from UEA.

**Results.** As illustrated in Fig. 3, UniTSC consistently outperforms all baselines, though a gap remains relative to the full-dataset upper bound. This reflects the inherent limitation imposed by the extreme scarcity of samples, which constrains the precision of decision boundaries. Neverthe-

*Table 4.* The F1-Score Comparison in Anomaly Detection (red: best, blue: second). See Table 13 in **Appendix D.7** for full results.

| Models | Random | K-Center | Herding | DC | MTT | DM | CondTSC | TimeDC | **UniTSC** | *Full* | *TimesNet* | *GPT4TS* |
|---|---|---|---|---|---|---|---|---|---|---|---|---|
| SMAP | 64.06 | 63.94 | 64.11 | 65.39 | 65.40 | 65.31 | 66.13 | 65.84 | **67.27** | *68.95* | *69.39* | *72.88* |
| PSM | 80.97 | 83.57 | 83.59 | 89.39 | 88.44 | 90.32 | 94.34 | 94.13 | **95.83** | *96.37* | *97.34* | *97.13* |
| SMD | 78.20 | 79.60 | 79.21 | 80.72 | 80.21 | 80.61 | 81.80 | 80.24 | **83.35** | *85.17* | *84.61* | *86.89* |
| SWaT | 79.41 | 79.32 | 79.52 | 81.57 | 81.46 | 82.07 | 85.59 | 85.25 | **85.66** | *92.16* | *93.02* | *94.23* |
| MSL | 71.00 | 71.14 | 71.05 | 74.12 | 73.20 | 74.62 | 75.92 | 74.75 | **76.03** | *79.56* | *81.84* | *82.45* |
| **Average** | 74.73 | 75.51 | 75.50 | 78.24 | 77.74 | 78.59 | 80.75 | 80.05 | **81.63** | *84.44* | *85.24* | *86.72* |

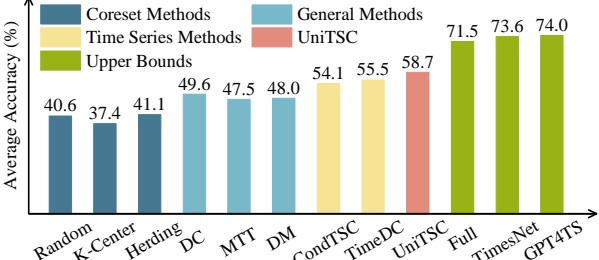

*Figure 3.* Average Accuracy Comparison in Classification. See Table 12 in **Appendix D.7** for full results.

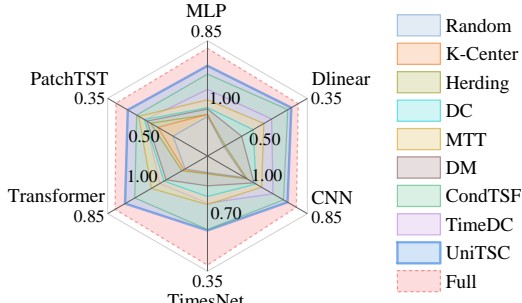

*Figure 4.* Cross-Architecture MSE Performance Comparison on ETTh1 Forecasting. See Table 14 in **Appendix D.7** for full results.

less, UniTSC demonstrates that sample quality can outweigh quantity: by maximizing the information content of synthetic samples, it effectively preserves discriminative features. Compared to coreset selection methods, UniTSC achieves an average accuracy improvement of up to 21.3%, indicating that the 5 synthesized samples serve as highly condensed class prototypes. These prototypes capture critical class-specific patterns more faithfully than raw selections, thereby maximizing separability within a limited budget.

### 5.5. Anomaly Detection Task

**Setups.** We set the condensed dataset size $M$ to 100 samples, aligning with the input window length.

**Results.** As shown in Table 4, UniTSC significantly outperforms all baselines. Despite relying on only 100 condensed samples, our method retains, on average, 96.7% of the performance of full-data training. Remarkably, on the SMD dataset, UniTSC achieves 97.9% performance retention while using merely 0.01% of the original training data. These findings suggest that such anomaly detection datasets exhibit high redundancy, dominated by repetitive normal patterns that contribute little marginal value to boundary definition. By capturing both spectral information and topological structures, UniTSC effectively condenses these dynamics into compact normality prototypes. This enables the establishment of precise and robust boundaries for distinguishing anomalies, even under extreme data compression.

### 5.6. Cross-Architecture Evaluation

To assess the generalizability of the condensed data, we conduct cross-architecture evaluations on representative forecasting datasets (ETTh1 and Electricity). We test

three architectural paradigms—MLP, CNN, and Transformer—selecting state-of-the-art representatives for each: DLinear (Zeng et al., 2023), TimesNet (Wu et al., 2023), and PatchTST (Nie et al., 2023). As shown in Fig. 4, UniTSC consistently achieves the lowest MSE across diverse architectures, demonstrating its ability to mitigate proxy-specific inductive biases. These results indicate that the synthetic data generated by UniTSC are not merely overfit to a specific proxy model but serve as architecture-agnostic training signals. They can be effectively utilized by MLPs, CNNs, and Transformers alike, confirming that UniTSC captures the intrinsic generative mechanisms of time series rather than structural artifacts of any single neural network.

### 5.7. Additional Studies in Appendix

We further evaluate UniTSC across six key dimensions:

**(i) Ablation Study.** Removing individual modules leads to notable performance degradation, confirming that each component is critical for capturing the underlying physical generative mechanisms (**Appendix D.1**).

**(ii) Sensitivity.** We analyze the sensitivity of the balancing weights, with results exhibiting consistent performance and strong robustness (**Appendix D.2**).

**(iii) Efficiency.** UniTSC enables up to a $10\times$ downstream model training speedup while achieving over 99.2% storage compression, demonstrating remarkable efficiency for resource-constrained scenarios (**Appendix D.3**).

**(iv) Condensation Ratios.** UniTSC maintains SOTA results with consistent gains across different condensation budgets, validating its robustness to sample scale (**Appendix D.4**).

**(v) Optimization Dynamics.** Loss evolution reveals a clear three-stage coarse-to-fine convergence pattern, highlighting the effectiveness of the tri-space alignment (**Appendix D.5**).

**(vi) Visualization Case Studies.** Visual evaluations across temporal, spectral, and topological views confirm that UniTSC effectively captures the underlying generative mechanisms (**Appendix D.6**).

## 6. Limitations

While UniTSC demonstrates strong generalization and data efficiency, we acknowledge the following limitations:

**Computational Overhead.** To ensure universal applicability, tri-space alignment introduces higher upfront computational costs during offline condensation compared to task-specific baselines. However, this one-time investment is effectively amortized by substantial recurring benefits in downstream training acceleration (see **Appendix D.3**).

**Classification Boundary Coverage.** While a single-batch budget suffices for sequence-to-sequence tasks where generative mechanisms are densely packed, it is less effective for classification tasks with high intra-class variance (see Table 12). In such scenarios, a strictly minimal budget cannot fully cover intricate decision boundaries, requiring a scaled sample budget to capture sufficient distributional diversity.

## 7. Conclusion

In this paper, we propose UniTSC, the first unified dataset condensation framework for general time series analysis. By integrating a Multi-View Hybrid Encoder with a Tri-Space Alignment Paradigm, UniTSC systematically captures temporal, spectral, and topological characteristics of time series, addressing the theoretical limitation that naive parameter-space alignment alone fails to preserve high-frequency details and multivariate dependencies, and thereby providing a principled framework for time-series dataset condensation. This work provides a versatile and efficient foundation for future research in data-centric temporal modeling.

## Acknowledgements

This work was supported in part by the NSFC under Grants No. (62402422 and U23A20296), Ningbo Natural Science Foundation under Grant No. 2025J007, Zhejiang Provincial Natural Science Foundation of China under Grant No. LZ25F020001, Yongjiang Talent Introduction Programme (2024A-162-G), and Zhejiang Province's 'Lingyan' R&D Project under Grant No. 2024C01259. Ziquan Fang is the corresponding author.

## Impact Statement

This paper presents work whose goal is to advance the field of Machine Learning. There are many potential societal consequences of our work, none of which we feel must be specifically highlighted here.

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

# Appendix

In the subsequent sections, we present supplementary materials to provide more details, offering deeper insights, theoretical backing, and extensive experimental results for readers seeking further clarification. The appendix is organized as follows:

In **Section A**, we provide comprehensive reviews of the related literature, covering both general time series analysis methods and the development of dataset condensation techniques.

In **Section B**, we present the methodology details, including (i) a summary of frequently used notations, (ii) the theoretical proof of the insufficiency of parameter space alignment, (iii) the pseudo code for the condensation procedure, and (iv) the theoretical proof of the generalization error bound.

In **Section C**, we describe the implementation details to ensure reproducibility, covering (i) statistical specifications of the 22 datasets across four tasks, (ii) detailed descriptions of the nine baselines and two state-of-the-art upper bound models, (iii) precise definitions of evaluation metrics, and (iv) specific experimental settings.

In **Section D**, we present additional experimental details to further demonstrate the superiority and robustness of UniTSC. This includes (i) comprehensive ablation studies confirming the critical role of each component, (ii) a hyperparameter sensitivity analysis verifying the robustness of the balancing weights, (iii) four experimental analyses demonstrating advantages in condensation, training, storage and data efficiency, (iv) robustness studies under varying condensation ratios, (v) an analysis of optimization dynamics verifying the coarse-to-fine convergence pattern, (vi) qualitative visualizations of the synthetic data from temporal, spectral, and topological perspectives, and (vii) the full tabulated results for all downstream tasks and cross-architecture evaluation.

Collectively, the appendix provides theoretical justification, empirical validation, and qualitative evidence that UniTSC preserves the underlying generative mechanisms of time series for general time series analysis.

## A. Related Work

### A.1. General Time Series Analysis

Time series analysis covers a wide range of tasks, such as forecasting (Qiu et al., 2026), classification (Liu et al., 2025a), imputation (Lai et al., 2024), and anomaly detection (Shao et al., 2025). While early research focused on task-specific architectures (Nie et al., 2023; Liu et al., 2024a; Wang et al., 2024), recent studies have shifted focus from single-task optimization to foundation models, utilizing large-scale representation learning to extract highly transferable features (Wu et al., 2023; Zhou et al., 2023; Goswami et al., 2024). However, this model-centric universal paradigm often comes with substantial computational costs: as data scale and task diversity explode, the burdens of model training, storage, and hyperparameter tuning grow non-linearly (Wang et al., 2025). Consequently, the core challenge has transitioned from merely pursuing structural generality to addressing efficiency bottlenecks at the data level (Das et al., 2024), calling for methods that achieve transferable universal capabilities with significantly reduced training costs.

### A.2. Dataset Condensation and Distillation

Dataset condensation (or distillation) aims to synthesize a small yet informative dataset, enabling neural networks to achieve comparable performance to full-data training. Early approaches often relied on coreset selection (Agarwal et al., 2004; Feldman et al., 2020), which filters representative samples based on heuristic rules (Lucic et al., 2017; Chai et al., 2023) or geometric distances (Har-Peled & Kushal, 2005; Feldman & Langberg, 2011). Recently, the synthesis-matching distillation paradigm has become mainstream, optimizing synthetic data by minimizing discrepancies in gradients (Zhao et al., 2021; Kim et al., 2022), feature distributions (Zhao et al., 2023; Zhao & Bilen, 2023), or training trajectories (Cazenavette et al., 2022; Cui et al., 2023). Additionally, some methods leverage kernel ridge regression (Nguyen et al., 2021a; Lee et al., 2022) to simplify this bi-level optimization problem into a single-level one. While effective, these methods primarily target image or graph data and are difficult to adapt directly to time series, characterized by temporal dynamics, complex dependencies, and sequential structures (Qiu et al., 2024). Naive transplantation of image-based techniques often fails to capture these intricate temporal evolutions, resulting in the loss of critical features and limiting the utility of the condensed data.

# B. Methodology Details

## B.1. Notations

Table 5 summarizes the frequently used notations and their definitions used throughout this paper.

*Table 5.* Summary of Notations

| Notation | Description |
|---|---|
| $\mathbf{X}$ | Time series sequence of observations $\{\mathbf{x}_1, \ldots, \mathbf{x}_L\}$ |
| $L$ | Length of the input sequence |
| $C$ | Number of channels (variables) |
| $\mathcal{T}$ | Real (full) training dataset |
| $\mathcal{S}$ | Synthetic (condensed) dataset |
| $N$ | Number of samples in real dataset $\mathcal{T}$ |
| $M$ | Number of samples in synthetic dataset $\mathcal{S}$ (condensed budget) |
| $\mathcal{X}_{syn}$ | Latent synthetic tensor |
| $\theta^*$ | Parameters of the expert model |
| $\hat{\theta}$ | Parameters of the student model |
| $T$ | Number of matching steps (trajectory length) |

## B.2. Proof of Proposition 4.1

Proposition 4.1 formalizes a widely observed empirical phenomenon in time-series condensation: parameter-space matching alone preferentially preserves low-frequency components while failing to constrain high-frequency dynamics and inter-variable dependency structures. Under Neural Tangent Kernel (NTK) dynamics (Rahaman et al., 2019), this bias is inherent and cannot be eliminated without introducing explicit spectral and topological regularization.

*Proof.* We analyze the insufficiency of parameter space alignment by mapping the parameter optimization dynamics to the function space, decomposing the error into spectral and topological components.

**Frequency Domain Insufficiency.** Updating the synthetic data $\mathcal{S}$ by matching the parameter trajectories can be modeled as a meta-optimization problem. According to the NTK theory, the evolution of a neural network during gradient descent approximates linear dynamics in the kernel space.

Let $H(x, x')$ be the NTK of the network, with the eigen-expansion $H(x, x') = \sum_k \lambda_k \phi_k(x)\phi_k(x')$, where $\lambda_k$ are eigenvalues and $\{\phi_k\}$ are orthonormal eigenfunctions. For standard neural network architectures applied to time series, the spectral bias principle (Rahaman et al., 2019) states that eigenvalues $\lambda_k$ decay significantly as the frequency $k$ increases (typically $\lambda_k \propto k^{-\alpha}$).

Let $\epsilon_k(t)$ denote the error projection on the $k$-th frequency component at optimization step $t$. The dynamics of the error under gradient descent with learning rate $\eta$ follow:

$$\frac{d}{dt}\epsilon_k(t) = -\eta\lambda_k\epsilon_k(t) \implies \epsilon_k(t) = e^{-\eta\lambda_k t}\epsilon_k(0). \tag{11}$$

In dataset condensation, the inner loop optimization steps $T$ are typically small (e.g., $T \in [10, 50]$). For high-frequency components (large $k$), $\lambda_k \to 0$, leading to $e^{-\eta\lambda_k T} \approx 1$.

Consequently, the parameter updates $\theta_T - \theta_0$ contain negligible information regarding high-frequency features. Since $\mathcal{L}_{param}$ relies on matching these updates, it fails to provide sufficient gradient signals to inject the high-frequency information of $\mathcal{T}$ into $\mathcal{S}$, resulting in a persistent spectral error.

**Topology Domain Insufficiency.** While parameter alignment constrains the model to learn similar decision boundaries, recovering the full topological structure of a multivariate time series $\mathbf{X} \in \mathbb{R}^{L \times C}$ is an ill-posed inverse problem. The topological structure is characterized by second-order statistics $\Sigma = \mathbb{E}[\mathbf{X}\mathbf{X}^T]$.

Given that the synthetic budget $M \ll N$, the problem is under-determined. The solution space $\Omega_{sol} = \{\mathcal{S} \mid \mathcal{L}_{param}(\mathcal{S}) < \delta\}$ is vast. Crucially, the scalar objective $\mathcal{L}_{param}$ is **largely insensitive** to transformations that preserve the decision boundary but alter the internal covariance structure. Consequently, the gradient $\nabla_{\mathcal{S}}\mathcal{L}_{param}$ provides **negligible** driving force

---

**Algorithm 1** The Condensation Procedure of UniTSC via Tri-Space Alignment

---

**Input:** real dataset $\mathcal{T}$, trajectory matching steps $T$, learning rate $\eta$, balancing weights $w_1, w_2, w_3$, number of experts $N_{expert}$, expert train epochs $E_{expert}$, condensation epochs $E_{syn}$
**Output:** synthetic dataset $\mathcal{S}$

1  Initialize latent synthetic parameters $\mathcal{X}_{syn}$ based on weak task priors      // Initialization (see Sec. 4.1)
2  **for** $i = 1$ **to** $N_{expert}$ **do**
3      Initialize $\theta_0^{*(i)}$ randomly
4      Train expert model on $\mathcal{T}$ and record trajectory buffer $\mathcal{B} \leftarrow \mathcal{B} \cup \{\theta_t^{*(i)}\}_{t=0}^{E_{expert}}$      // Collect expert dynamics
5  **for** *epoch* $e = 1$ **to** $E_{syn}$ **do**
6      Sample expert trajectory segment $(\theta_t^*, \theta_{t+T}^*) \sim \mathcal{B}$
7      Initialize student model: $\hat{\theta}_0 \leftarrow \theta_t^*$      // Sync start state
8      Construct synthetic batch $\mathcal{S} = \text{Gen}(\mathcal{X}_{syn})$      // Generate synthetic data (see Sec. 4.1)
9      **for** *step* $t = 0$ **to** $T - 1$ **do**
10        Forward student $\hat{\theta}_t$ on $\mathcal{S}$ using Multi-View Hybrid Encoder      // Extract multi-view features (see Sec. 4.2)
11        Update student: $\hat{\theta}_{t+1} \leftarrow \hat{\theta}_t - \eta \nabla_{\hat{\theta}_t} \mathcal{L}_{\text{train}}(\mathcal{S}; \hat{\theta}_t)$      // Inner-loop optimization
12     $\mathcal{L}_{param} = \frac{||\hat{\theta}_T - \theta_{t+T}^*||_2^2}{||\theta_t^* - \theta_{t+T}^*||_2^2}$      // Eq. 5
13     Compute frequency and topological alignment losses $\mathcal{L}_{freq}, \mathcal{L}_{topo}$      // Eq. 6 & Eq. 7
14     Calculate total loss: $\mathcal{L}_{total} = w_1 \mathcal{L}_{param} + w_2 \mathcal{L}_{freq} + w_3 \mathcal{L}_{topo}$      // Eq. 9
15     Update latent parameters: $\mathcal{X}_{syn} \leftarrow \mathcal{X}_{syn} - \nabla_{\mathcal{X}_{syn}} \mathcal{L}_{total}$
16 **return** optimized synthetic dataset $\mathcal{S}$ generated from $\mathcal{X}_{syn}$

---

to align $\Sigma_{\mathcal{S}}$ with $\Sigma_{\mathcal{T}}$. Therefore, unless initialized with the exact target topology (a zero-probability event), the optimization will converge to a solution where the topological mismatch remains strictly positive:

$$\|\Sigma_{\mathcal{S}} - \Sigma_{\mathcal{T}}\|_F^2 \geq \xi_{topo} > 0. \tag{12}$$

**Conclusion.** The derivations above demonstrate that the gradient flow of $\mathcal{L}_{param}$ is inherently limited by the student model's inductive biases. Specifically, since the gradient $\nabla_{\mathcal{S}} \mathcal{L}_{param}$ is back-propagated through a network subject to spectral bias, the gradient components corresponding to high-frequency and topological features are systematically suppressed (i.e., vanishing gradients). Consequently, even when accumulated over the entire condensation procedure, the total update magnitude in these domains remains negligible compared to the initialization discrepancy (i.e., $\|\Delta \mathcal{S}_{high}\| \ll \|\mathcal{S}_{init} - \mathcal{T}\|_{high}$). The final distributional discrepancy is thus dominated by the uncorrected initialization errors:

$$\mathcal{D}(\mathcal{S}, \mathcal{T}) \geq C_1 \|\mathcal{F}(\mathcal{S})_{high} - \mathcal{F}(\mathcal{T})_{high}\|^2 + C_2 \|\Sigma_{\mathcal{S}} - \Sigma_{\mathcal{T}}\|_F^2. \tag{13}$$

This concludes the proof that parameter space alignment alone is insufficient, as it fails to generate the necessary gradient flux to drive high-frequency and topological features away from their random initialization states. □

### B.3. Algorithm of Condensation

In this section, we provide the detailed optimization procedure of UniTSC. The framework follows a bi-level optimization paradigm. Specifically, the outer loop optimizes the synthetic latent tensor $\mathcal{X}_{syn}$ by minimizing the tri-space alignment loss, while the inner loop emulates the training process of a student model on the synthetic data to align its trajectory with pre-trained expert trajectories. The complete algorithm is outlined in Alg. 1.

### B.4. Proof of Theorem 4.2

We provide the detailed proof for the generalization error bound. We aim to demonstrate that minimizing the proposed tri-space alignment objectives explicitly reduces the upper bound of the generalization gap between the student model trained on synthetic data $\mathcal{S}$ and the expert model trained on real data $\mathcal{T}$.

*Proof.* Let $\theta_t^*$ and $\hat{\theta}_t$ denote the parameters of the expert model and the student model at training step $t$, respectively. Let $\mathcal{R}(\theta)$ denote the expected risk. We adopt two standard assumptions regarding the loss landscape: (i) $G$-**Lipschitz**

**Continuity:** $|\mathcal{R}(\theta_1) - \mathcal{R}(\theta_2)| \leq G\|\theta_1 - \theta_2\|_2$; (ii) $\beta$**-Smoothness:** $\|\nabla\mathcal{L}(\theta_1) - \nabla\mathcal{L}(\theta_2)\|_2 \leq \beta\|\theta_1 - \theta_2\|_2$. Based on Lipschitz continuity, the generalization gap is bounded by the parameter trajectory divergence:

$$\mathcal{R}(\hat{\theta}_T) - \mathcal{R}(\theta_T^*) \leq G\|\hat{\theta}_T - \theta_T^*\|_2. \tag{14}$$

Consider the SGD update rules with learning rate $\eta$: $\theta_{t+1}^* = \theta_t^* - \eta\nabla\mathcal{L}(\mathcal{T}; \theta_t^*)$, and $\hat{\theta}_{t+1} = \hat{\theta}_t - \eta\nabla\mathcal{L}(\mathcal{S}; \hat{\theta}_t)$. Let $\epsilon_t = \|\hat{\theta}_t - \theta_t^*\|_2$ denote the parameter error at step $t$. The divergence at step $t + 1$ evolves as follows:

$$\epsilon_{t+1} = \|(\hat{\theta}_t - \theta_t^*) - \eta(\nabla\mathcal{L}(\mathcal{S}; \hat{\theta}_t) - \nabla\mathcal{L}(\mathcal{T}; \theta_t^*))\|_2. \tag{15}$$

Applying the triangle inequality yields:

$$\epsilon_{t+1} \leq \epsilon_t + \eta\|\nabla\mathcal{L}(\mathcal{S}; \hat{\theta}_t) - \nabla\mathcal{L}(\mathcal{T}; \theta_t^*)\|_2. \tag{16}$$

We decompose the gradient discrepancy term by introducing $\nabla\mathcal{L}(\mathcal{T}; \hat{\theta}_t)$:

$$\|\nabla\mathcal{L}(\mathcal{S}; \hat{\theta}_t) - \nabla\mathcal{L}(\mathcal{T}; \theta_t^*)\|_2 \leq \underbrace{\|\nabla\mathcal{L}(\mathcal{T}; \hat{\theta}_t) - \nabla\mathcal{L}(\mathcal{T}; \theta_t^*)\|_2}_{\text{Model Divergence}} + \underbrace{\|\nabla\mathcal{L}(\mathcal{S}; \hat{\theta}_t) - \nabla\mathcal{L}(\mathcal{T}; \hat{\theta}_t)\|_2}_{\text{Gradient Estimation Error } (\mathcal{E}_{align}^{(t)})}. \tag{17}$$

Using $\beta$-smoothness assumption, the Model Divergence is bounded by $\beta\|\hat{\theta}_t - \theta_t^*\|_2 = \beta\epsilon_t$. Thus, the recursion simplifies to:

$$\epsilon_{t+1} \leq (1 + \eta\beta)\epsilon_t + \eta\mathcal{E}_{align}^{(t)}. \tag{18}$$

Next, we bound the Gradient Estimation Error, denoted as $\mathcal{E}_{align}^{(t)} = \|\nabla\mathcal{L}(\mathcal{S}; \hat{\theta}_t) - \nabla\mathcal{L}(\mathcal{T}; \hat{\theta}_t)\|_2$, which quantifies the discrepancy between gradients computed on synthetic versus real data at the same parameter state. To maintain generality while establishing a connection between our proposed objectives and the generalization bound, we introduce the following structural consistency assumption:

**Assumption A.1 (Tri-Space Gradient Consistency)**. Positing that gradient deviations stem from representational mismatches in the underlying feature spaces, we assume that $\mathcal{E}_{align}^{(t)}$ is upper-bounded by the structural divergence across three domains. Formally, there exist non-negative constants $\lambda_1$, $\lambda_2$, $\lambda_3$ such that:

$$\mathcal{E}_{align}^{(t)} \leq \lambda_1\sqrt{\mathcal{L}_{param}} + \lambda_2\sqrt{\mathcal{L}_{freq}} + \lambda_3\sqrt{\mathcal{L}_{topo}}. \tag{19}$$

**Justification**: Assumption A.1 is theoretically grounded in decomposing gradient error into three complementary subspaces. Specifically: (i) **First-order Dynamics**: Under the assumption of $\beta$-smoothness, parameter distance strictly bounds gradient divergence. Thus, minimizing $\mathcal{L}_{param}$ acts as a direct proxy for minimizing the Euclidean error of the gradient vectors. (ii) **Spectral Regularization**: By leveraging Parseval's theorem (Oppenheim, 1999), $\mathcal{L}_{freq}$ explicitly bounds the spectral norm of the error matrix, thereby preventing gradient errors from concentrating in high-frequency bands that spatial losses often overlook. (iii) **Manifold Constraint**: $\mathcal{L}_{topo}$ acts as a structural regularizer that constrains the student's gradient flow to reside within the valid tangent space of the expert's implicit dependency graph, eliminating spurious orthogonal components. Consequently, the weighted sum in Assumption A.1 provides a holistic upper bound on gradient error by enforcing consistency across magnitude, spectrum, and topology.

Finally, solving the recurrence relation in Eq. (18) from $t = 0$ to $T - 1$, assuming shared initialization ($\epsilon_0 = 0$):

$$\epsilon_T \leq \sum_{t=0}^{T-1}(1 + \eta\beta)^{T-1-t}\eta\mathcal{E}_{align}^{(t)}. \tag{20}$$

Substituting this result into Eq. (14) and expanding $\mathcal{E}_{align}^{(t)}$, we obtain the final bound:

$$\mathcal{R}(\hat{\theta}_T) - \mathcal{R}(\theta_T^*) \leq G\eta\sum_{t=0}^{T-1}(1 + \eta\beta)^{T-1-t}\left(\lambda_1\sqrt{\mathcal{L}_{param}} + \lambda_2\sqrt{\mathcal{L}_{freq}} + \lambda_3\sqrt{\mathcal{L}_{topo}}\right). \tag{21}$$

$$\square$$

This establishes that the generalization gap is strictly bounded by the accumulated tri-space alignment errors. Therefore, minimizing the alignment objective effectively lowers the worst-case upper bound of the generalization error.

# C. Implementation Details

## C.1. Datasets

To comprehensively evaluate the generalization capability of our framework, we conduct experiments on a wide range of real-world datasets across four mainstream time series analysis tasks: forecasting, imputation, classification, and anomaly detection. These datasets cover diverse domains including energy, weather, healthcare, transportation, and server monitoring. Detailed statistics of all utilized datasets are summarized in Table 6.

**Dataset Selection.** We select benchmarks tailored to each task according to Wu et al. (2023) and Zhou et al. (2023). For **Forecasting** and **Imputation**, we employ widely used benchmarks such as ETT subsets (Zhou et al., 2021), Weather (Wetterstation), Electricity (Trindade), and Traffic (PeMS), which vary significantly in temporal dependencies and variable correlations. For **Classification**, we select 10 datasets from the UEA archive (Bagnall et al., 2018), ranging from low-dimensional gesture data to high-dimensional transport metrics. For **Anomaly Detection**, we utilize five widely used datasets, SMD (Su et al., 2019), MSL (Hundman et al., 2018), SMAP (Hundman et al., 2018), SWaT (Mathur & Tippenhauer, 2016) and PSM (Abdulaal et al., 2021), characterized by high dimensions and large sample sizes.

**Condensation Protocol.** To ensure a consistent evaluation protocol, we align the condensed dataset size with the standard input configurations of downstream tasks. Specifically, the number of condensed samples is set to 96 for forecasting and imputation, and 100 for anomaly detection, matching the standard input sequence lengths used in these domains. For classification, we adopt a budget of 5 samples per class. Finally, we report the *Condensation Ratio* as shown in Table 6, defined as the size of the condensed set divided by the original training set size. It is worth noting that for classification, the condensation ratio is determined by the specific goal of validating whether a minimal budget (i.e., 5 prototypes) effectively captures class-wise dynamics, prioritizing the evaluation of information density over raw storage reduction.

*Table 6.* Dataset Descriptions

| Tasks | Datasets | Dim | Length | Train | Test | Information | Condensation Ratio |
|---|---|---|---|---|---|---|---|
| Forecasting | ETTm1, ETTm2 | 7 | 96 | 34,369 | 11,425 | Electricity | 0.28% |
| | ETTh1, ETTh2 | 7 | 96 | 8,449 | 2,785 | Electricity | 1.14% |
| | Weather | 21 | 96 | 36,696 | 10,444 | Weather | 0.26% |
| | Electricity | 321 | 96 | 18,221 | 5,165 | Electricity | 0.53% |
| | Traffic | 862 | 96 | 12,089 | 3,413 | Transportation | 0.79% |
| Imputation | ETTm1, ETTm2 | 7 | 96 | 34,369 | 11,425 | Electricity | 0.28% |
| | ETTh1, ETTh2 | 7 | 96 | 8,449 | 2,785 | Electricity | 1.14% |
| | Weather | 21 | 96 | 36,696 | 10,444 | Weather | 0.26% |
| | Electricity | 321 | 96 | 18,221 | 5,165 | Electricity | 0.53% |
| Classification (UEA) | EthanolConcentration | 3 | 1751 | 261 | 263 | Alcohol | 7.66% |
| | FaceDetection | 144 | 62 | 5890 | 3524 | Face | 0.17% |
| | Handwriting | 3 | 152 | 150 | 850 | Handwriting | 86.67% |
| | Heartbeat | 61 | 405 | 204 | 205 | Heart Beat | 4.90% |
| | JapaneseVowels | 12 | 29 | 270 | 370 | Voice | 16.67% |
| | PEMS-SF | 963 | 144 | 267 | 173 | Transport | 13.11% |
| | SelfRegulationSCP1 | 6 | 896 | 268 | 293 | Health | 3.73% |
| | SelfRegulationSCP2 | 7 | 1152 | 200 | 180 | Health | 5.00% |
| | SpokenArabicDigits | 13 | 93 | 6599 | 2199 | Voice | 0.76% |
| | UWaveGestureLibrary | 3 | 315 | 120 | 320 | Gesture | 33.33% |
| Anomaly Detection | SMAP | 25 | 100 | 135,183 | 427,617 | Spacecraft | 0.07% |
| | PSM | 25 | 100 | 132,481 | 87,841 | Server | 0.08% |
| | SMD | 38 | 100 | 708,405 | 708,420 | Server | 0.01% |
| | SWaT | 51 | 100 | 495,000 | 449,919 | Infrastructure | 0.02% |
| | MSL | 55 | 100 | 73,729 | 58,317 | Spacecraft | 0.14% |

**Experimental Settings.** We standardize the evaluation protocols as follows: (i) for **forecasting**, we use a fixed past sequence length of 96 to predict horizons of $\{96, 192, 336\}$; (ii) for **imputation**, to evaluate capacity under different missing rates, we randomly mask time points at ratios of $\{12.5\%, 25\%, 37.5\%, 50\%\}$; (iii) for **classification**, following the variable-length preprocessing in (Zerveas et al., 2021), we perform sequence-level classification to verify high-level representation learning; (iv) for **anomaly detection**, following (Xu et al., 2022), we split datasets into non-overlapping segments via sliding windows and utilize reconstruction error as anomaly criterion.

### C.2. Baselines

In this section, we provide detailed descriptions of nine baseline methods and two state-of-the-art general models used for performance upper bound evaluation. The baselines are categorized into three groups: coreset selection strategies, general dataset condensation methods, and time series condensation methods.

**Coreset Selection Strategies.** These methods select a representative subset of the original training data without synthesizing new samples. They serve as fundamental baselines to validate the necessity of synthesis-based condensation.

- **Random Selection:** A naive baseline that uniformly samples a fixed number of instances from the original training set to form the core set. It serves as the performance lower bound.

- **K-Center (Farahani & Hekmatfar, 2009):** A geometry-based selection method. It performs K-Center clustering on the original dataset and samples from the cluster centers to construct the core set.

- **Herding (Welling, 2009):** A greedy iterative strategy. At each step, it adds a sample that minimizes the Euclidean distance between the mean vector of the current subset and the global mean of the full dataset.

**General Dataset Condensation Methods.** These methods are originally designed for image data but are adapted here for time series by employing specialized 1D backbones to preserve temporal dependencies.

- **DC (Zhao et al., 2021):** A gradient matching method. It synthesizes a small set of informative samples by minimizing the distance between the model parameter gradients computed on the real dataset and those on the synthetic dataset.

- **MTT (Cazenavette et al., 2022):** An advanced optimization method based on trajectory matching. It aligns the training trajectories of models trained on synthetic versus real data, effectively capturing long-term training dynamics.

- **DM (Zhao & Bilen, 2023):** A distribution matching method. It bypasses bi-level optimization by directly matching the feature distribution statistics of real and synthetic data within a random embedding space, achieving high efficiency.

**Time Series Condensation Methods.** These are specialized dataset condensation methods explicitly tailored to capture the temporal dependencies inherent in time series data.

- **CondTSF (Ding et al., 2024):** A method tailored for time series forecasting. It introduces a lightweight plugin module that decomposes the optimization objective into gradient and numerical terms to match prediction errors.

- **CondTSC (Liu et al., 2024b):** A method designed for time series classification. It utilizes Fourier transforms and matches training dynamics in both time and frequency domains to capture periodic patterns.

- **TimeDC (Miao et al., 2024):** A method for forecasting and classification tasks. It combines decomposition-driven frequency matching with curriculum training trajectory matching to preserve temporal dependencies.

Given the task-specific nature of existing time-series condensation methods, we employ CondTSF and TimeDC for forecasting, and CondTSC and TimeDC for classification. While existing time-series dataset condensation methods are intrinsically designed with task-specific biases (e.g., forecasting or classification), we adapt CondTSF and TimeDC to the imputation task and anomaly detection task to ensure a fair comparison.

**Full Dataset Upper Bounds.** To quantify the performance gap between our condensed data and the state-of-the-art results achievable on the full dataset, we employ two representative advanced methods as reference upper bounds:

- **TimesNet (Wu et al., 2023):** A leading general model for time series analysis. Employing modular multi-periodicity analysis, it transforms 1D time series into 2D tensors to effectively capture temporal variations and periodic patterns.

- **GPT4TS (Zhou et al., 2023):** A novel paradigm that uses LLMs for time series. By freezing the GPT-2 and fine-tuning only a minimal set of parameters, it activates the universal feature representation capabilities for time series data.

## C.3. Metrics

To comprehensively evaluate the performance of our framework across different tasks, we employ standard metrics widely adopted in the time series analysis community (Wu et al., 2023; Zhou et al., 2023; Goswami et al., 2024).

For forecasting and imputation tasks, we adopt Mean Squared Error (MSE) and Mean Absolute Error (MAE):

$$\text{MSE} = \frac{1}{n} \sum_{i=1}^{n} (y_i - \hat{y}_i)^2, \quad \text{MAE} = \frac{1}{n} \sum_{i=1}^{n} |y_i - \hat{y}_i|, \tag{22}$$

where $y_i$ and $\hat{y}_i$ denote the ground truth and predicted values, and $n$ represents the total number of evaluation samples.

For classification, we utilize Accuracy as the primary metric:

$$\text{Accuracy} = \frac{1}{n} \sum_{i=1}^{n} \mathbb{I}(c_i = \hat{c}_i), \tag{23}$$

where $c_i$ and $\hat{c}_i$ correspond to the actual and predicted class labels, and $\mathbb{I}(\cdot)$ denotes the indicator function.

For anomaly detection, we report Precision, Recall, and the F1-score derived from the confusion matrix:

$$\text{Precision} = \frac{TP}{TP + FP}, \quad \text{Recall} = \frac{TP}{TP + FN}, \quad \text{F1} = \frac{2 \cdot \text{Precision} \cdot \text{Recall}}{\text{Precision} + \text{Recall}}, \tag{24}$$

where $TP$, $FP$, and $FN$ represent the number of True Positives, False Positives, and False Negatives, respectively. To ensure statistical reliability, all experiments are repeated three times, and the mean results are reported.

## C.4. Implementations

All experiments are conducted on a server running Rocky Linux 8.8, equipped with NVIDIA A40 GPUs. We implement UniTSC using Python 3.9.2 and PyTorch 2.5.1. For the backbone architecture, we employ a 3-layer encoder with 4 self-attention heads. For the multi-scale design, the coarse patch length is set to 16 with a stride of 8, while the fine-grained patch length is set to half that size (i.e., 8). Additionally, the frequency selection parameter $k$ was set to 5. To construct the expert replay buffers, we generate 10 independent expert trajectories using an SGD optimizer, where each trajectory consists of 50 training epochs. For the tri-space alignment objective, we set the balancing weights to $w_1 = 1$, $w_2 = 0.001$, and $w_3 = 100$. Specifically, $w_1$ is set to 1 to establish parameter matching as the primary optimization backbone, while $w_2$ and $w_3$ are calibrated to scale the magnitudes of $\mathcal{L}_{freq}$ and $\mathcal{L}_{topo}$, ensuring they function as effective regularizers without dominating the gradient flow. During distillation, the synthetic data is updated with a learning rate of 0.1. Further implementation details and source code are available in the code repository: https://github.com/ZJU-DAILY/UniTSC.

# D. Experimental Details

## D.1. Ablation Study

To rigorously verify the contribution of each component within the UniTSC framework, we conduct a comprehensive ablation study on forecasting tasks. We design five distinct variants to isolate specific modules of the Multi-View Hybrid Encoder and the Tri-Space Alignment objectives: (i) w/o Patching, which omits the patching mechanism; (ii) w/o Multi-Scale, which relies solely on single-scale patching; (iii) w/o Freq, which excludes the frequency mix block and frequency loss; (iv) w/o Topo, which discards the topological branch and topology loss; and (v) w/o Freq&Topo, which relies exclusively on the parameter space to guide the condensation process. The quantitative results are detailed in Table 7, and the optimization dynamics are illustrated in Fig. 5.

As shown in Table 7, the experimental results reveal critical insights: (i) The w/o Patching variant exhibits the most severe performance degradation. This underscores that simple point-wise mapping is inadequate for capturing the local semantic contexts intrinsic to time series representation. (ii) Similarly, the significant degradation in the w/o Multi-Scale variant validates that a single temporal resolution is insufficient to simultaneously capture coarse-grained global trends and fine-grained local fluctuations. (iii) While w/o Freq and w/o Topo variants exhibit relatively moderate declines individually, this indicates that the frequency and topological modules serve as critical physical regularizers that refine the synthesis of

*Table 7.* Ablation Study on Different Components on Forecasting Task (**black**: best)

| Variants | w/o Patching | | w/o Multi-Scale | | w/o Freq | | w/o Topo | | w/o Freq&Topo | | **UniTSC** | |
| --- | --- | --- | --- | --- | --- | --- | --- | --- | --- | --- | --- | --- |
| Metric | MAE | MSE | MAE | MSE | MAE | MSE | MAE | MSE | MAE | MSE | MAE | MSE |
| ETTh1 | 0.433 | 0.428 | 0.429 | 0.424 | 0.420 | 0.416 | 0.419 | 0.416 | 0.442 | 0.436 | **0.415** | **0.413** |
| ETTh2 | 0.374 | 0.335 | 0.364 | 0.327 | 0.364 | 0.326 | 0.358 | 0.322 | 0.385 | 0.343 | **0.357** | **0.320** |
| ETTm1 | 0.424 | 0.412 | 0.417 | 0.406 | 0.413 | 0.403 | 0.405 | 0.392 | 0.430 | 0.418 | **0.398** | **0.381** |
| ETTm2 | 0.304 | 0.213 | 0.298 | 0.210 | 0.293 | 0.207 | 0.287 | 0.203 | 0.324 | 0.221 | **0.280** | **0.200** |
| Weather | 0.258 | 0.229 | 0.254 | 0.225 | 0.249 | 0.215 | 0.244 | 0.207 | 0.267 | 0.241 | **0.240** | **0.202** |
| Electricity | 0.334 | 0.244 | 0.332 | 0.244 | 0.330 | 0.242 | 0.327 | 0.240 | 0.342 | 0.253 | **0.322** | **0.236** |
| Traffic | 0.490 | 0.772 | 0.488 | 0.767 | 0.473 | 0.754 | 0.469 | 0.744 | 0.504 | 0.783 | **0.460** | **0.709** |

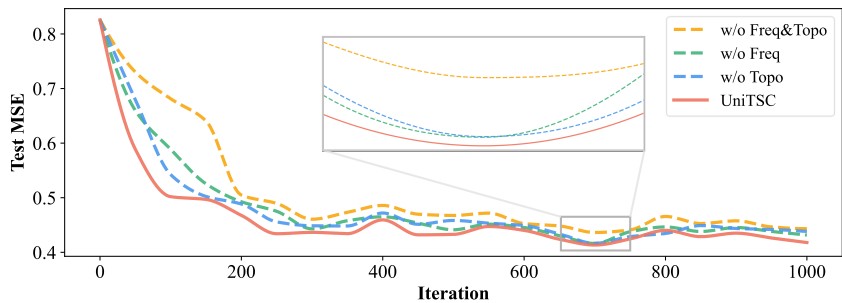

*Figure 5.* Comparison of MSE Convergence with Different Loss Components on ETTh1

periodic laws and multivariate correlations. (iv) Notably, the w/o Freq&Topo variant suffers the most significant degradation. This result empirically supports our theoretical hypothesis (Proposition 4.1) that parameter matching alone is insufficient. Without the explicit guidance of frequency and topological constraints, the optimization degenerates into numerical matching that fails to recover the authentic physical generative mechanisms of the data.

We further analyze the training stability by tracking the test MSE evolution on ETTh1, as shown in Fig. 5. The UniTSC (red line) exhibits a robust coarse-to-fine convergence pattern, rapidly and stably reaching the lowest error floor. In contrast, the w/o Freq&Topo variant (yellow line) exhibits the poorest performance, characterized by significant volatility and a premature plateau. Meanwhile, the w/o Freq (green line) and w/o Topo (blue line) variants track the full model most closely, yet still exhibit a discernible convergence lag during the initial training phase. These observations confirm that the Tri-Space Alignment paradigm effectively constrains the optimization trajectory to a valid manifold, preventing the model from drifting into spurious local minima or structural degeneration often encountered in single-view optimization.

### D.2. Hyperparameter Sensitivity Study

To verify the robustness of our Tri-Space Alignment Paradigm and validate the choice of balancing weights, we conduct a sensitivity analysis on forecasting task. We specifically focus on the weights for frequency alignment ($w_2$) and topology alignment ($w_3$), while fixing the parameter alignment weight ($w_1 = 1$) as the optimization backbone.

**Sensitivity to Frequency Weight** ($w_2$). We vary $w_2$ across the magnitude range $\{10^{-4}, \ldots, 10^{-2}\}$ while keeping $w_3 = 100$. As shown in Fig. 6(a), the model demonstrates high stability across most datasets, with a consistent performance optimum

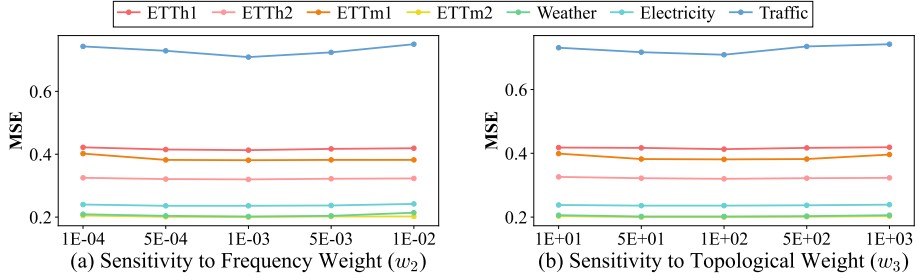

*Figure 6.* Hyperparameter Sensitivity Analysis of the Frequency Weight ($w_2$) and Topological Weight ($w_3$)

observed around $w_2 = 10^{-3}$. Notably, for complex datasets like Traffic (blue line), the performance exhibits a discernible U-shaped trend. Specifically, when $w_2$ is too small ($< 5 \times 10^{-4}$), the constraint is insufficient to capture high-frequency details. Conversely, increasing $w_2$ beyond $5 \times 10^{-3}$ leads to MSE degradation, validating our hypothesis that excessive frequency weighting forces the model to overfit spectral noise at the expense of the main forecasting objective.

**Sensitivity to Topology Weight** ($w_3$). Similarly, we evaluate $w_3$ within the range $\{10^1, \dots, 10^3\}$ with $w_2$ fixed at $10^{-3}$. The results in Fig. 6(b) demonstrate that UniTSC is remarkably robust to $w_3$ variations, with performance curves remaining relatively flat. However, an optimal configuration is identified at $w_3 = 10^2$, where the lowest MSE is consistently achieved across all datasets. Lower weights fail to enforce sufficient structural alignment, resulting in weak inter-variable coupling, while excessively high weights ($w_3 = 10^3$) lead to slight performance drops in sensitive datasets (e.g., Traffic), attributable to optimization instability where the topological gradient dominates the learning signal.

### D.3. Efficiency Study

To demonstrate the practical feasibility of our framework, we conduct a comprehensive efficiency study on forecasting task from four perspectives: the computational cost of the condensation process (Fig. 7), the training efficiency for downstream applications (Fig. 8(a)), the storage efficiency (Fig. 8(b)), and the data efficiency (Table 8).

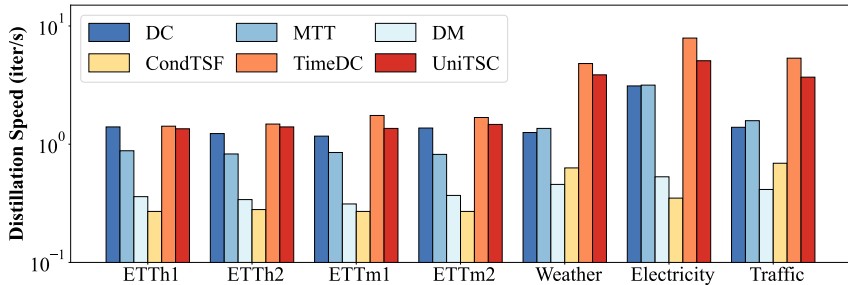

*Figure 7.* Computational Efficiency of the Condensation Process

**Condensation Efficiency.** We first evaluate the computational overhead required to synthesize the condensed data. As illustrated in Fig. 7, comparisons of distillation speed (iter/s) reveal that UniTSC incurs a moderate overhead relative to baselines. This is an expected consequence of our design: unlike naive parameter matching, UniTSC performs rigorous alignment across three orthogonal spaces (parameter, frequency, and topology) to preserve underlying generative mechanisms. We argue that this additional computation is a justifiable trade-off for ensuring universality. Crucially, given that dataset condensation is a one-time offline process, this marginal upfront cost is effectively amortized by the substantial long-term benefits, specifically the superior downstream generalization and significant training acceleration.

**Training Efficiency.** The core value of dataset condensation is to reduce the resource consumption for downstream model training. As shown in Fig. 8(a), models trained on condensed data achieve an acceleration of approximately $10\times$ compared to the full dataset. Specifically, consider the Electricity dataset as an example: the condensation process takes $\sim$5,080s, yet it enables a net saving of $\sim$45,050s for a single training run compared to the full dataset. In real-world scenarios requiring repeated training, such as hyperparameter tuning or neural architecture search, these time savings scale linearly with the number of training iterations. Thus, condensation is a one-time offline cost, whereas the downstream benefits are recurring.

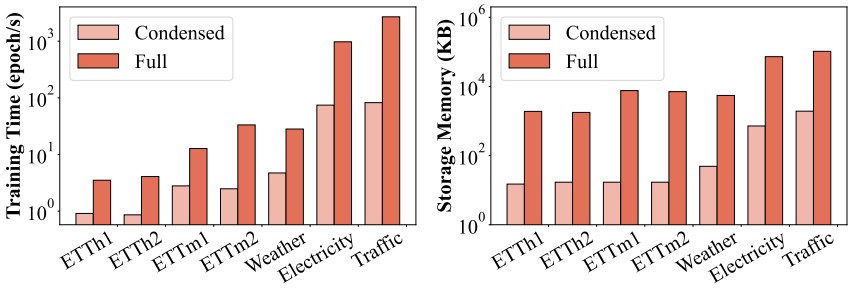

*Figure 8.* Downstream Model Training Efficiency and Storage Efficiency

**Storage Efficiency.** Beyond computational speed, minimizing memory footprint is critical for deployment in extreme edge environments (e.g., IoT sensors, embedded MCUs). As shown in Fig. 8(b), the condensed data reduces storage requirements

*Table 8.* MSE Performance Comparison with Few-Shot Random Sampling (red: best, blue: second, *italics*: upper bound)

| Dataset | ETTh1 | | | | | | ETTh2 | | | | | | ETTm1 | | | | | |
|---|---|---|---|---|---|---|---|---|---|---|---|---|---|---|---|---|---|---|
| Category | UniTSC | Few-Shot | | | | Full | UniTSC | Few-Shot | | | | Full | UniTSC | Few-Shot | | | | Full |
| Ratio | 1.14% | 1% | 5% | 10% | 20% | 100% | 1.14% | 1% | 5% | 10% | 20% | 100% | 0.28% | 1% | 5% | 10% | 20% | 100% |
| DLinear | **0.414** | 0.686 | 0.627 | 0.570 | 0.504 | *0.386* | **0.355** | 0.360 | 0.359 | 0.356 | **0.355** | *0.333* | **0.386** | 0.629 | 0.510 | 0.455 | 0.416 | *0.345* |
| TimesNet | **0.576** | 0.896 | 0.817 | 0.771 | 0.733 | *0.384* | **0.354** | 0.422 | 0.400 | 0.386 | 0.373 | *0.340* | 0.764 | 0.739 | 0.692 | 0.588 | **0.462** | *0.338* |
| PatchTST | **0.418** | 0.877 | 0.751 | 0.643 | 0.544 | *0.378* | **0.326** | 0.405 | 0.388 | 0.371 | 0.351 | *0.302* | **0.367** | 0.722 | 0.515 | 0.444 | 0.403 | *0.329* |

| Model | ETTm2 | | | | | | Weather | | | | | | Electricity | | | | | |
|---|---|---|---|---|---|---|---|---|---|---|---|---|---|---|---|---|---|---|
| Category | UniTSC | Few-Shot | | | | Full | UniTSC | Few-Shot | | | | Full | UniTSC | Few-Shot | | | | Full |
| Ratio | 0.28% | 1% | 5% | 10% | 20% | 100% | 0.26% | 1% | 5% | 10% | 20% | 100% | 0.53% | 1% | 5% | 10% | 20% | 100% |
| DLinear | 0.236 | 0.234 | 0.233 | 0.225 | **0.217** | *0.193* | 0.237 | 0.214 | 0.211 | 0.206 | **0.202** | *0.196* | 0.290 | 0.654 | 0.337 | 0.285 | **0.257** | *0.197* |
| TimesNet | **0.231** | 0.265 | 0.248 | 0.241 | 0.235 | *0.187* | 0.223 | 0.245 | 0.223 | 0.217 | **0.208** | *0.172* | **0.258** | 1.059 | 0.937 | 0.853 | 0.417 | *0.168* |
| PatchTST | **0.198** | 0.252 | 0.233 | 0.219 | 0.206 | *0.175* | 0.208 | 0.234 | 0.213 | 0.209 | **0.204** | *0.177* | **0.215** | 0.490 | 0.313 | 0.281 | 0.258 | *0.181* |

from the $10^4$ KB scale to a mere $10^2$ KB scale, achieving a compression ratio of up to 99.2%. While the absolute size of standard benchmarks may appear manageable, this compression capability is a critical enabler for large-scale distributed sensor networks, where storing and transmitting high-frequency data from millions of endpoints incurs prohibitive costs.

**Data Efficiency.** We evaluate the information density of the condensed data by comparing UniTSC against a Random Selection baseline under varying data budgets (1%, 5%, 10%, 20%). To ensure robustness, we test diverse architectures using state-of-the-art models: DLinear (MLP-based), TimesNet (CNN-based), and PatchTST (Transformer-based). As detailed in Table 8, models trained on UniTSC's condensed data (often utilizing $< 1\%$ of the training set) frequently outperform the same models trained on 20% of randomly selected raw data. This result highlights the exceptional information density and strong cross-architecture generalization of our synthetic samples. It confirms that UniTSC successfully distills the essence of the dataset, empirically validating our assertion that one batch is enough to capture the core physical mechanisms.

Although UniTSC requires a one-time moderate computational investment during the condensation phase, it yields substantial long-term benefits. By enabling models to be trained orders of magnitude faster with minimal storage memory, UniTSC establishes a highly efficient condense-once, train-many paradigm for general time series analysis.

### D.4. Condensation Ratios Study

Given the inherent sensitivity of classification tasks to dataset size, we evaluate the impact of the condensation ratio, defined as the number of samples per class ($spc$). We extend the standard setting ($spc = 5$) to include extreme compression ($spc = 1$) and higher-budget ($spc = 10$) scenarios. The results are shown in Table 9. Note that we exclude the Handwriting dataset from the $spc = 10$ experiments, as its limited training set size of 150 samples is insufficient for this setting.

Experimental results demonstrate that UniTSC consistently achieves state-of-the-art performance across the vast majority of scenarios, exhibiting strong robustness. These findings further validate that the proposed Multi-View Encoder and Tri-Space Alignment effectively enable UniTSC to distill the most critical discriminative prototypes, even under the extreme constraint of using only one sample per class.

Moreover, by analyzing the accuracy evolution, we identify two distinct performance scaling: (i) For datasets exhibiting unimodal characteristics (e.g., FaceDetection and SelfRegulationSCP2), a single condensed sample is sufficient to capture the core discriminative patterns. Consequently, performance saturates early, yielding only marginal accuracy gains as the sample budget increases. (ii) For more complex datasets characterized by high intra-class variance (e.g., JapaneseVowels and PEMS-SF), a single sample fails to cover the underlying data manifold fully. In such cases, increasing the sample budget introduces necessary distributional diversity, significantly expanding the model's generalization boundary.

### D.5. Optimization Dynamics Study

To validate the stability and efficacy of our Tri-Space Alignment Paradigm, we track the evolution of training losses and test performance (MSE) on the ETTh1 forecasting task. As illustrated in Fig. 9, the optimization process exhibits a clear coarse-to-fine convergence pattern, distinctively divided into three phases. The observed coarse-to-fine convergence pattern aligns well with the design of the tri-space alignment paradigm: parameter alignment dominates early-stage coarse fitting,

*Table 9.* Performance Comparison with Different Condensation Ratios (red: best, blue: second, *italics*: upper bound)

| Models
Datasets | spc | Random | K-Center | Herding | DC | MTT | DM | CondTSC | TimeDC | **UniTSC** | *Full* | *TimesNet* | *GPT4TS* |
|---|---|---|---|---|---|---|---|---|---|---|---|---|---|
| EthanolConcentration | 1 | 18.3 | 16.3 | 17.5 | 21.7 | 23.6 | 25.8 | 23.6 | 25.9 | **27.8** | *31.2* | *35.7* | *34.2* |
|  | 5 | 20.2 | 18.6 | 21.7 | 23.2 | 25.1 | 26.4 | 26.2 | 26.6 | **28.9** |  |  |  |
|  | 10 | 23.2 | 21.3 | 23.6 | 25.9 | 25.9 | 27.5 | 29.4 | 28.1 | **29.7** |  |  |  |
| FaceDetection | 1 | 49.3 | 49.2 | 47.3 | 19.4 | 18.3 | 21.4 | 50.4 | 50.8 | **51.7** | *65.6* | *68.6* | *69.2* |
|  | 5 | 49.2 | 49.3 | 49.6 | 33.6 | 27.9 | 38.2 | 51.8 | 52.5 | **53.2** |  |  |  |
|  | 10 | 49.2 | 50.0 | 48.7 | 49.7 | 46.7 | 50.1 | 53.0 | 53.7 | **56.0** |  |  |  |
| Handwriting | 1 | 11.2 | 10.1 | 11.9 | 12.7 | 12.2 | 15.1 | 14.6 | 15.9 | **18.5** | *31.6* | *32.1* | *32.7* |
|  | 5 | 15.8 | 15.5 | 16.9 | 19.1 | 18.4 | 19.4 | 22.3 | 21.8 | **25.8** |  |  |  |
|  | 10 | - | - | - | - | - | - | - | - | **-** |  |  |  |
| Heartbeat | 1 | 43.9 | 34.2 | 47.3 | 56.1 | 55.1 | 51.2 | 61.9 | 68.8 | **71.7** | *75.6* | *78.0* | *77.2* |
|  | 5 | 49.3 | 44.9 | 50.2 | 59.5 | 59.0 | 51.6 | 63.6 | 71.2 | **73.2** |  |  |  |
|  | 10 | 53.2 | 53.2 | 55.1 | 62.9 | 59.5 | 54.0 | 72.1 | 73.7 | **74.2** |  |  |  |
| JapaneseVowels | 1 | 32.2 | 38.6 | 37.6 | 51.1 | 51.9 | 45.1 | 51.4 | 48.9 | **54.3** | *94.9* | *98.4* | *98.6* |
|  | 5 | 45.9 | 47.0 | 43.5 | 52.7 | 53.5 | 47.6 | 54.1 | 52.7 | **57.3** |  |  |  |
|  | 10 | 49.7 | 51.9 | 50.3 | 55.1 | 55.4 | 50.3 | 62.2 | 59.7 | **68.9** |  |  |  |
| PEMS-SF | 1 | 27.2 | 17.3 | 26.0 | 40.5 | 39.3 | 38.7 | 41.3 | 49.7 | 45.7 | *86.0* | *89.6* | *87.9* |
|  | 5 | 41.6 | 20.2 | 38.7 | 49.7 | 42.2 | 45.1 | 52.1 | 52.6 | **57.5** |  |  |  |
|  | 10 | 49.7 | 48.0 | 56.1 | 55.5 | 51.4 | 54.9 | 61.9 | 56.1 | **66.2** |  |  |  |
| SelfRegulationSCP1 | 1 | 67.6 | 71.7 | 61.1 | 66.9 | 64.5 | 63.5 | 78.8 | 77.5 | **87.0** | *91.8* | *91.8* | *93.2* |
|  | 5 | 70.6 | 72.7 | 64.5 | 72.4 | 71.0 | 70.6 | 81.6 | 79.9 | **87.7** |  |  |  |
|  | 10 | 73.7 | 74.7 | 71.0 | 77.8 | 76.8 | 79.9 | 84.3 | 82.3 | **88.4** |  |  |  |
| SelfRegulationSCP2 | 1 | 42.2 | 46.7 | 48.3 | 48.9 | 51.1 | 50.1 | 50.0 | 49.4 | **57.2** | *58.9* | *57.2* | *59.4* |
|  | 5 | 43.3 | 50.0 | 49.4 | 51.7 | 52.2 | 50.8 | 52.4 | 51.7 | **57.8** |  |  |  |
|  | 10 | 43.3 | 50.6 | 52.2 | 52.8 | 52.8 | 53.3 | 55.7 | 53.9 | **58.3** |  |  |  |
| SpokenArabicDigits | 1 | 16.5 | 14.3 | 20.6 | 56.7 | 49.1 | 54.9 | 53.1 | 70.1 | 59.4 | *97.1* | *99.0* | *99.2* |
|  | 5 | 25.8 | 24.2 | 30.2 | 64.0 | 58.5 | 61.6 | 64.6 | 74.8 | 72.9 |  |  |  |
|  | 10 | 36.0 | 46.1 | 45.3 | 71.1 | 66.3 | 68.1 | 80.9 | 80.0 | **84.4** |  |  |  |
| UWaveGestureLibrary | 1 | 32.5 | 22.2 | 36.6 | 66.9 | 64.7 | 64.4 | 69.1 | 68.4 | **71.6** | *82.2* | *85.3* | *88.1* |
|  | 5 | 44.1 | 31.6 | 46.6 | 70.6 | 67.5 | 69.1 | 72.8 | 71.3 | **73.1** |  |  |  |
|  | 10 | 54.1 | 41.9 | 57.5 | 73.1 | 70.9 | 73.4 | 76.9 | 75.3 | **78.4** |  |  |  |
| Avg |  | 40.6 | 39.0 | 42.2 | 50.4 | 48.6 | 49.0 | 55.6 | 56.7 | **59.9** | *71.5* | *73.6* | *74.0* |

while frequency and topological constraints progressively refine the solution toward physically consistent manifolds.

**Phase I: Coarse Alignment** (Iterations 0-200). In the early iterations, both the parameter loss (red line) and test MSE (green line) exhibit a rapid decline. This phase is dominated by trajectory matching, where the synthetic data guides the student model to follow the expert's parameter update direction. However, the frequency loss (blue line) still exhibits brief spikes and noticeable fluctuations, suggesting that the spectral characteristics remain unstable while the model is primarily learning the overall dynamics.

**Phase II: Refinement** (Iterations 200-700). In the mid-stage, once the coarse parameter backbone is established, the optimization enters a critical refinement phase. During this period, the frequency loss (blue line) and topology loss (purple line) become the primary drivers of performance improvement. While the parameter loss (red line) exhibits oscillatory behavior, the test MSE (green line) maintains a general downward trend. This phase empirically validates the necessity of the Tri-Space Alignment Paradigm: the continuous reduction in frequency and topology losses indicates that the synthetic data is being refined to capture high-frequency details (e.g., seasonality) and complex inter-variable dependencies that parameter matching alone fails to model.

**Phase III: Oscillation & Stabilization** (Iterations 700-1000). In the final stage, the optimization reaches a state of dynamic

*Figure 9.* Evolution of Training Objectives and MSE Performance on ETTh1

equilibrium. Although the loss values exhibit minor fluctuations, they remain consistently bounded within a low range, while the test MSE stabilizes at its optimal plateau. This stability indicates that the synthetic dataset has achieved maximum information density, successfully condensing the core information of the original dataset into the condensed budget, while simultaneously preventing solution degeneration through sustained topological constraints.

Overall, these optimization dynamics confirm that UniTSC follows a robust coarse-to-fine learning trajectory. By effectively balancing global trend matching with local spectral-topological refinement, our framework synthesizes highly informative samples while avoiding the spectral bias and structural degeneration that are often observed in single-objective optimization.

### D.6. Visualization of Synthetic Data

To intuitively assess the quality of the condensed data, we conduct a visual evaluation on the ETTh1 dataset. As illustrated in Fig. 10, we compare the temporal evolution against three representative baselines (Random, MTT, and CondTSF). Furthermore, we conduct a detailed comparative analysis of spectral consistency (Fig. 11) and multivariate topology (Fig. 12) specifically against the best-performing baseline CondTSF.

**Temporal Evolution.** As shown in Fig. 10, distinct performance gaps are observable across the methods. Although Random selection preserves the inherent temporal order of raw sequences, it functions merely as a stochastic snapshot, lacking the prototypical patterns required for effective condensation. Similarly, MTT exhibits artifacts resembling averaged noise, which fails to preserve local geometries and leads to a loss of distinct peak-valley structures. While CondTSF successfully reproduces global seasonality, it suffers from significant over-smoothing. It effectively behaves as a low-pass filter, retaining fundamental frequencies but eroding the high-frequency volatility and local textures essential for real-world time series. In contrast, UniTSC achieves an optimal balance between global coherence and local fidelity, preserving not only the global periodicity but also the sharpness of peaks and the intricacy of local fluctuations.

**Spectral Consistency.** To validate the generative mechanisms in the frequency domain, we compare the Power Spectral Density (PSD) as shown in Fig. 11. For CondTSF, the spectral energy of the prediction horizon is significantly attenuated compared to the look-back window, accompanied by a notable deviation in shape. This inconsistency indicates a failure to maintain energy distribution during generation. Conversely, UniTSC demonstrates superior preservation of the spectral structure. Its prediction horizon tracks the look-back window closely, maintaining consistent peak positions and fluctuation shapes with only marginal amplitude reduction, thereby ensuring spectral fidelity.

**Multivariate Topology.** As shown in Fig. 12, we examine the inter-variable covariance structures using correlation matrix heatmaps. The results reveal that CondTSF distorts the multivariate topology by introducing spurious artifacts. Specifically, it generates strong negative correlations where none exist (e.g., between HUFL and HULL) and artificially amplifies weak correlations (e.g., between MULL and LUFL). Conversely, UniTSC successfully preserves the intrinsic block structures and variable groupings. Despite minor deviations in the OT variable, UniTSC maintains the dominant positive correlations and avoids extreme structural errors, exhibiting high fidelity to the ground truth multivariate topology.

In conclusion, these qualitative results confirm that UniTSC generates richly informative condensed representations, rather than simple mean matching. By leveraging the Multi-View Encoder and Tri-Space Alignment, UniTSC reconstructs the complex structural characteristics of the original data. This ensures that the synthetic samples possess not only accurate temporal dynamics but also the same spectral density and informative multivariate relationships as the real data distribution.

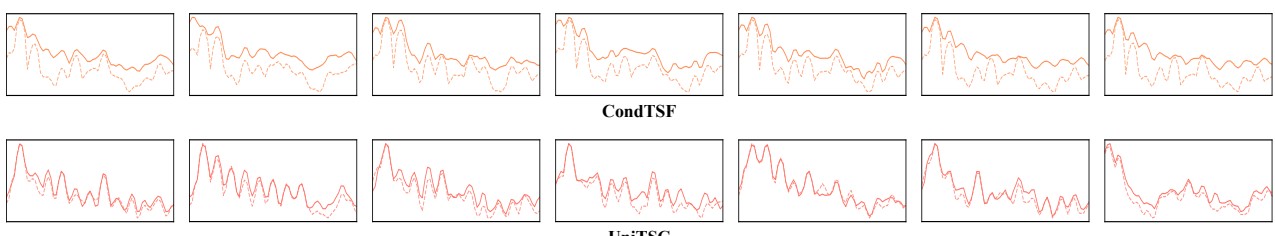

*Figure 10.* Temporal Visualization of Synthetic Data on ETTh1 (Solid: Look-back Window, Dashed: Prediction Horizon)

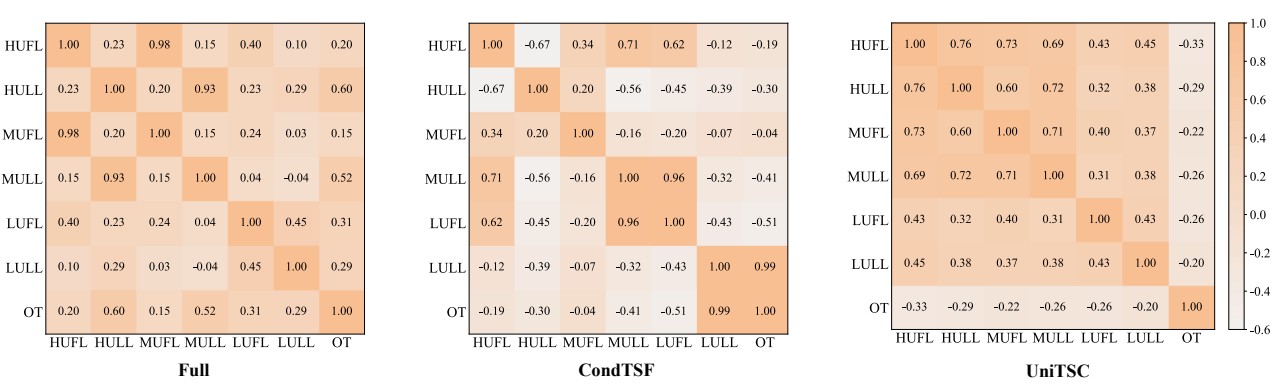

*Figure 11.* Spectral Visualization of Synthetic Data on ETTh1 (Solid: Look-back Window, Dashed: Prediction Horizon)

*Figure 12.* Topological Visualization of Synthetic Data on ETTh1

## D.7. Full Results

Due to the space limitation of the main text, we provide detailed results for all experiments in the following: forecasting task in Table 10, imputation task in Table 11, classification task in Table 12, anomaly detection task in Table 13, and cross-architecture evaluation in Table 14.

**Forecasting Task.** Table 10 presents a comprehensive evaluation across seven benchmark datasets and three prediction horizons $\{96, 192, 336\}$. UniTSC consistently outperforms all baselines, achieving the lowest MSE and MAE errors across all settings. Notably, as the prediction horizon extends from 96 to 336, UniTSC exhibits superior robustness, maintaining a significant performance advantage. This suggests that the proposed tri-space alignment effectively preserves global periodic patterns with topological structures, thereby mitigating the cumulative error problem typically encountered in long-sequence forecasting. Remarkably, despite using only a single batch-equivalent budget (96 samples), UniTSC achieves performance highly comparable to state-of-the-art models trained on full datasets (e.g., TimesNet and GPT4TS), demonstrating that a single high-quality condensed batch is sufficient to capture the essential generative mechanisms of time-series data.

**Imputation Task.** Table 11 details imputation performance across varying masking ratios ranging from 12.5% to 50%. While reconstruction errors naturally increase with higher missing rates, UniTSC consistently maintains the lowest error rates among all condensation methods, demonstrating superior resilience to data sparsity. On several datasets, UniTSC reduces the average error by approximately 41% compared to the second-best method. This improvement indicates that the Tri-Space Alignment effectively reconstructs local textures and inter-variable dependencies, enabling UniTSC to preserve fine-grained information essential for reconstruction even in high-masking scenarios.

**Classification Task.** Table 12 reports classification accuracy on 10 UEA datasets with a 5-shot budget. UniTSC consistently achieves the highest average accuracy, outperforming state-of-the-art baselines. This advantage is particularly pronounced on complex datasets such as PEMS-SF, indicating the successful distillation of informative class prototypes that capture discriminative features more effectively than random selection. While prioritizing physical fidelity for universality may be less direct than purely discriminative strategies in rare instances (e.g., SpokenArabicDigits), this deliberate trade-off ensures superior overall robustness. Compared to the coreset methods, UniTSC yields a significant improvement of up to 21.3%, demonstrating its ability to maximize information density within a minimal budget.

**Anomaly Detection Task.** Table 13 shows the Precision, Recall, and F1-Scores across five large-scale anomaly detection benchmarks. UniTSC achieves a dominant performance, retaining an average of 96.7% of the full-data F1-Score using only 100 condensed samples. While baselines like MTT may occasionally exhibit higher Recall, this often stems from their selection of noisy samples that fail to adequately represent normality, leading to poor reconstruction of normal data and a surge in False Positives. In contrast, UniTSC consistently achieves the optimal F1-Score, validating its superior ability to distill complex dynamics into compact normality prototypes that define precise and robust anomaly boundaries.

**Cross-Architecture Evaluation.** Table 14 evaluates the generalization capability of condensed data across MLP-based, CNN-based, and Transformer-based architectures. The results indicate that synthetic data generated by UniTSC consistently yields the lowest MSE and MAE errors across all evaluated backbones. These findings suggest that the condensed data is not overfitted to the specific proxy model used during synthesis. Indeed, UniTSC maintains exceptional performance on Transformer-based models like PatchTST while transferring effectively to structurally different architectures such as DLinear and TimesNet. This evidence supports our core argument that Tri-Space Alignment enables the capture of intrinsic generative mechanisms rather than merely memorizing the structural artifacts of a single neural network.

*Table 10.* Full Performance Comparison in Forecasting (red: best, blue: second, *italics*: upper bound)

| Models | | Random | | K-Center | | Herding | | DC | | MTT | | DM | | CondTSF | | TimeDC | | UniTSC | | Full | | TimesNet | | GPT4TS | |
|---|---|---|---|---|---|---|---|---|---|---|---|---|---|---|---|---|---|---|---|---|---|---|---|---|---|
| Metric | | MAE | MSE | MAE | MSE | MAE | MSE | MAE | MSE | MAE | MSE | MAE | MSE | MAE | MSE | MAE | MSE | MAE | MSE | *MAE* | *MSE* | *MAE* | *MSE* | *MAE* | *MSE* |
| ETTh1 | 96 | 0.580 | 0.761 | 0.580 | 0.749 | 0.569 | 0.720 | 0.527 | 0.589 | 0.505 | 0.561 | 0.552 | 0.657 | 0.448 | 0.451 | 0.464 | 0.484 | **0.415** | **0.413** | *0.405* | *0.397* | *0.402* | *0.384* | *0.397* | *0.376* |
| | 192 | 0.617 | 0.764 | 0.619 | 0.764 | 0.620 | 0.777 | 0.548 | 0.646 | 0.535 | 0.618 | 0.588 | 0.672 | 0.487 | 0.523 | 0.493 | 0.532 | **0.445** | **0.481** | *0.440* | *0.455* | *0.429* | *0.436* | *0.418* | *0.416* |
| | 336 | 0.644 | 0.797 | 0.640 | 0.781 | 0.660 | 0.804 | 0.571 | 0.698 | 0.563 | 0.668 | 0.624 | 0.715 | 0.524 | 0.583 | 0.524 | 0.586 | **0.466** | **0.530** | *0.466* | *0.503* | *0.469* | *0.491* | *0.433* | *0.442* |
| | Avg | 0.614 | 0.774 | 0.613 | 0.765 | 0.616 | 0.767 | 0.549 | 0.644 | 0.534 | 0.616 | 0.588 | 0.681 | 0.486 | 0.519 | 0.494 | 0.534 | **0.442** | **0.475** | *0.437* | *0.452* | *0.433* | *0.437* | *0.416* | *0.411* |
| ETTh2 | 96 | 0.557 | 0.569 | 0.515 | 0.497 | 0.516 | 0.496 | 0.467 | 0.489 | 0.458 | 0.479 | 0.468 | 0.496 | 0.385 | 0.340 | 0.369 | 0.337 | **0.357** | **0.320** | *0.345* | *0.298* | *0.374* | *0.340* | *0.342* | *0.285* |
| | 192 | 0.575 | 0.580 | 0.534 | 0.517 | 0.534 | 0.518 | 0.478 | 0.498 | 0.473 | 0.487 | 0.483 | 0.513 | 0.439 | 0.411 | 0.421 | 0.418 | **0.400** | **0.388** | *0.397* | *0.384* | *0.414* | *0.402* | *0.389* | *0.354* |
| | 336 | 0.597 | 0.604 | 0.565 | 0.538 | 0.552 | 0.532 | 0.493 | 0.507 | 0.490 | 0.503 | 0.537 | 0.534 | 0.482 | 0.471 | 0.466 | 0.454 | **0.435** | **0.433** | *0.433* | *0.426* | *0.452* | *0.452* | *0.407* | *0.373* |
| | Avg | 0.576 | 0.584 | 0.538 | 0.517 | 0.534 | 0.515 | 0.479 | 0.498 | 0.474 | 0.490 | 0.496 | 0.514 | 0.435 | 0.408 | 0.419 | 0.403 | **0.397** | **0.380** | *0.392* | *0.369* | *0.413* | *0.398* | *0.379* | *0.337* |
| ETTm1 | 96 | 0.548 | 0.712 | 0.551 | 0.709 | 0.543 | 0.692 | 0.487 | 0.510 | 0.474 | 0.462 | 0.490 | 0.514 | 0.450 | 0.412 | 0.493 | 0.442 | **0.398** | **0.381** | *0.367* | *0.339* | *0.375* | *0.338* | *0.346* | *0.292* |
| | 192 | 0.600 | 0.838 | 0.588 | 0.812 | 0.592 | 0.821 | 0.516 | 0.572 | 0.513 | 0.576 | 0.515 | 0.567 | 0.463 | 0.443 | 0.512 | 0.466 | **0.406** | **0.418** | *0.387* | *0.384* | *0.387* | *0.374* | *0.372* | *0.332* |
| | 336 | 0.606 | 0.845 | 0.607 | 0.847 | 0.629 | 0.911 | 0.548 | 0.635 | 0.546 | 0.638 | 0.528 | 0.627 | 0.489 | 0.476 | 0.532 | 0.483 | **0.413** | **0.434** | *0.410* | *0.420* | *0.411* | *0.410* | *0.394* | *0.366* |
| | Avg | 0.585 | 0.798 | 0.582 | 0.789 | 0.588 | 0.808 | 0.517 | 0.572 | 0.511 | 0.559 | 0.511 | 0.569 | 0.467 | 0.444 | 0.512 | 0.463 | **0.406** | **0.411** | *0.388* | *0.381* | *0.391* | *0.374* | *0.371* | *0.330* |
| ETTm2 | 96 | 0.478 | 0.427 | 0.430 | 0.397 | 0.426 | 0.393 | 0.331 | 0.271 | 0.338 | 0.277 | 0.336 | 0.275 | 0.319 | 0.251 | 0.325 | 0.279 | **0.280** | **0.200** | *0.265* | *0.180* | *0.267* | *0.187* | *0.262* | *0.173* |
| | 192 | 0.511 | 0.476 | 0.451 | 0.416 | 0.448 | 0.410 | 0.357 | 0.319 | 0.376 | 0.335 | 0.362 | 0.326 | 0.346 | 0.293 | 0.354 | 0.312 | **0.319** | **0.259** | *0.307* | *0.245* | *0.309* | *0.249* | *0.301* | *0.229* |
| | 336 | 0.541 | 0.530 | 0.471 | 0.440 | 0.469 | 0.438 | 0.384 | 0.360 | 0.404 | 0.388 | 0.399 | 0.378 | 0.375 | 0.341 | 0.385 | 0.357 | **0.352** | **0.315** | *0.346* | *0.305* | *0.351* | *0.321* | *0.341* | *0.286* |
| | Avg | 0.510 | 0.477 | 0.451 | 0.418 | 0.448 | 0.414 | 0.357 | 0.317 | 0.373 | 0.333 | 0.366 | 0.326 | 0.347 | 0.295 | 0.355 | 0.316 | **0.317** | **0.258** | *0.306* | *0.243* | *0.309* | *0.252* | *0.301* | *0.229* |
| Weather | 96 | 0.368 | 0.326 | 0.366 | 0.324 | 0.366 | 0.323 | 0.347 | 0.338 | 0.310 | 0.290 | 0.328 | 0.298 | 0.309 | 0.274 | 0.257 | 0.239 | **0.240** | **0.202** | *0.230* | *0.193* | *0.220* | *0.172* | *0.212* | *0.162* |
| | 192 | 0.410 | 0.382 | 0.406 | 0.377 | 0.408 | 0.379 | 0.369 | 0.371 | 0.353 | 0.365 | 0.344 | 0.335 | 0.336 | 0.301 | 0.295 | 0.275 | **0.280** | **0.255** | *0.268* | *0.240* | *0.261* | *0.219* | *0.248* | *0.204* |
| | 336 | 0.444 | 0.434 | 0.443 | 0.434 | 0.443 | 0.433 | 0.395 | 0.423 | 0.390 | 0.419 | 0.367 | 0.370 | 0.356 | 0.343 | 0.339 | 0.322 | **0.315** | **0.307** | *0.305* | *0.293* | *0.306* | *0.280* | *0.286* | *0.254* |
| | Avg | 0.407 | 0.381 | 0.405 | 0.378 | 0.406 | 0.378 | 0.370 | 0.377 | 0.351 | 0.358 | 0.346 | 0.334 | 0.334 | 0.306 | 0.297 | 0.279 | **0.278** | **0.255** | *0.268* | *0.242* | *0.262* | *0.224* | *0.249* | *0.207* |
| Electricity | 96 | 0.437 | 0.331 | 0.435 | 0.338 | 0.441 | 0.336 | 0.416 | 0.319 | 0.396 | 0.298 | 0.412 | 0.315 | 0.339 | 0.246 | 0.334 | 0.243 | **0.322** | **0.236** | *0.294* | *0.208* | *0.272* | *0.168* | *0.238* | *0.139* |
| | 192 | 0.454 | 0.356 | 0.457 | 0.358 | 0.457 | 0.359 | 0.431 | 0.334 | 0.408 | 0.311 | 0.424 | 0.322 | 0.363 | 0.269 | 0.367 | 0.271 | **0.341** | **0.253** | *0.303* | *0.217* | *0.289* | *0.184* | *0.251* | *0.153* |
| | 336 | 0.482 | 0.379 | 0.481 | 0.378 | 0.485 | 0.372 | 0.458 | 0.355 | 0.432 | 0.332 | 0.437 | 0.337 | 0.397 | 0.307 | 0.401 | 0.316 | **0.383** | **0.301** | *0.315* | *0.223* | *0.300* | *0.198* | *0.266* | *0.169* |
| | Avg | 0.458 | 0.355 | 0.457 | 0.358 | 0.461 | 0.356 | 0.435 | 0.336 | 0.412 | 0.314 | 0.424 | 0.325 | 0.366 | 0.274 | 0.367 | 0.277 | **0.348** | **0.264** | *0.304* | *0.216* | *0.287* | *0.183* | *0.252* | *0.154* |
| Traffic | 96 | 0.534 | 0.779 | 0.533 | 0.770 | 0.534 | 0.775 | 0.527 | 0.800 | 0.509 | 0.751 | 0.519 | 0.763 | 0.497 | 0.739 | 0.502 | 0.742 | **0.460** | **0.709** | *0.380* | *0.602* | *0.321* | *0.593* | *0.282* | *0.388* |
| | 192 | 0.538 | 0.828 | 0.542 | 0.840 | 0.539 | 0.836 | 0.535 | 0.817 | 0.516 | 0.760 | 0.528 | 0.794 | 0.511 | 0.773 | 0.526 | 0.797 | **0.468** | **0.740** | *0.382* | *0.614* | *0.336* | *0.617* | *0.290* | *0.407* |
| | 336 | 0.544 | 0.854 | 0.549 | 0.864 | 0.551 | 0.876 | 0.542 | 0.838 | 0.527 | 0.801 | 0.538 | 0.822 | 0.525 | 0.798 | 0.541 | 0.843 | **0.484** | **0.762** | *0.383* | *0.629* | *0.336* | *0.629* | *0.294* | *0.412* |
| | Avg | 0.539 | 0.820 | 0.541 | 0.825 | 0.541 | 0.829 | 0.535 | 0.818 | 0.517 | 0.771 | 0.528 | 0.793 | 0.511 | 0.770 | 0.523 | 0.794 | **0.470** | **0.737** | *0.382* | *0.615* | *0.331* | *0.613* | *0.289* | *0.402* |

*Table 11.* Full Performance Comparison in Imputation (red: best, blue: second, *italics*: upper bound)

| | Metric | Random MAE | Random MSE | K-Center MAE | K-Center MSE | Herding MAE | Herding MSE | DC MAE | DC MSE | MTT MAE | MTT MSE | DM MAE | DM MSE | CondTSF MAE | CondTSF MSE | TimeDC MAE | TimeDC MSE | UniTSC MAE | UniTSC MSE | Full MAE | Full MSE | TimesNet MAE | TimesNet MSE | GPT4TS MAE | GPT4TS MSE |
|---|---|---|---|---|---|---|---|---|---|---|---|---|---|---|---|---|---|---|---|---|---|---|---|---|---|
| ETTh1 | 12.5% | 0.429 | 0.410 | 0.419 | 0.407 | 0.403 | 0.366 | 0.321 | 0.244 | 0.295 | 0.208 | 0.370 | 0.302 | 0.262 | 0.174 | 0.277 | 0.182 | **0.238** | **0.131** | *0.212* | *0.102* | *0.159* | *0.057* | *0.140* | *0.043* |
| | 25.0% | 0.445 | 0.447 | 0.438 | 0.446 | 0.417 | 0.396 | 0.357 | 0.279 | 0.327 | 0.249 | 0.383 | 0.325 | 0.314 | 0.237 | 0.324 | 0.241 | **0.292** | **0.198** | *0.235* | *0.126* | *0.178* | *0.069* | *0.156* | *0.054* |
| | 37.5% | 0.467 | 0.492 | 0.461 | 0.498 | 0.433 | 0.432 | 0.379 | 0.319 | 0.350 | 0.276 | 0.398 | 0.348 | 0.323 | 0.246 | 0.341 | 0.257 | **0.312** | **0.226** | *0.257* | *0.151* | *0.196* | *0.084* | *0.180* | *0.072* |
| | 50.0% | 0.488 | 0.546 | 0.485 | 0.554 | 0.454 | 0.481 | 0.402 | 0.357 | 0.374 | 0.311 | 0.418 | 0.364 | 0.354 | 0.288 | 0.360 | 0.294 | **0.338** | **0.266** | *0.280* | *0.180* | *0.215* | *0.102* | *0.216* | *0.107* |
| | Avg | 0.457 | 0.474 | 0.451 | 0.476 | 0.427 | 0.419 | 0.365 | 0.300 | 0.336 | 0.261 | 0.392 | 0.335 | 0.313 | 0.236 | 0.326 | 0.244 | **0.295** | **0.205** | *0.246* | *0.140* | *0.187* | *0.078* | *0.173* | *0.069* |
| ETTh2 | 12.5% | 0.284 | 0.174 | 0.270 | 0.165 | 0.284 | 0.174 | 0.243 | 0.139 | 0.232 | 0.135 | 0.255 | 0.144 | 0.212 | 0.112 | 0.228 | 0.125 | **0.179** | **0.075** | *0.155* | *0.059* | *0.130* | *0.040* | *0.125* | *0.039* |
| | 25.0% | 0.320 | 0.202 | 0.285 | 0.182 | 0.289 | 0.182 | 0.261 | 0.153 | 0.252 | 0.148 | 0.273 | 0.168 | 0.238 | 0.134 | 0.246 | 0.139 | **0.181** | **0.078** | *0.163* | *0.065* | *0.141* | *0.046* | *0.135* | *0.044* |
| | 37.5% | 0.346 | 0.231 | 0.303 | 0.207 | 0.295 | 0.190 | 0.284 | 0.174 | 0.276 | 0.162 | 0.298 | 0.185 | 0.256 | 0.148 | 0.265 | 0.152 | **0.187** | **0.084** | *0.172* | *0.071* | *0.151* | *0.052* | *0.147* | *0.051* |
| | 50.0% | 0.373 | 0.260 | 0.321 | 0.222 | 0.301 | 0.198 | 0.302 | 0.198 | 0.295 | 0.186 | 0.314 | 0.203 | 0.269 | 0.163 | 0.287 | 0.178 | **0.195** | **0.091** | *0.181* | *0.078* | *0.162* | *0.060* | *0.158* | *0.059* |
| | Avg | 0.331 | 0.217 | 0.295 | 0.194 | 0.292 | 0.186 | 0.273 | 0.166 | 0.264 | 0.158 | 0.285 | 0.175 | 0.244 | 0.139 | 0.257 | 0.149 | **0.185** | **0.082** | *0.168* | *0.068* | *0.146* | *0.050* | *0.141* | *0.048* |
| ETTm1 | 12.5% | 0.312 | 0.237 | 0.304 | 0.223 | 0.313 | 0.239 | 0.285 | 0.198 | 0.289 | 0.202 | 0.276 | 0.185 | 0.256 | 0.153 | 0.261 | 0.157 | **0.217** | **0.118** | *0.138* | *0.046* | *0.101* | *0.023* | *0.085* | *0.017* |
| | 25.0% | 0.330 | 0.267 | 0.327 | 0.262 | 0.332 | 0.271 | 0.293 | 0.212 | 0.291 | 0.212 | 0.298 | 0.213 | 0.273 | 0.170 | 0.274 | 0.169 | **0.224** | **0.128** | *0.153* | *0.056* | *0.101* | *0.023* | *0.096* | *0.022* |
| | 37.5% | 0.351 | 0.306 | 0.352 | 0.309 | 0.352 | 0.308 | 0.304 | 0.224 | 0.302 | 0.223 | 0.313 | 0.240 | 0.287 | 0.197 | 0.295 | 0.206 | **0.230** | **0.132** | *0.166* | *0.066* | *0.111* | *0.029* | *0.111* | *0.029* |
| | 50.0% | 0.378 | 0.359 | 0.384 | 0.371 | 0.378 | 0.359 | 0.311 | 0.233 | 0.314 | 0.235 | 0.332 | 0.261 | 0.298 | 0.212 | 0.309 | 0.227 | **0.231** | **0.135** | *0.182* | *0.079* | *0.124* | *0.036* | *0.128* | *0.040* |
| | Avg | 0.343 | 0.292 | 0.342 | 0.291 | 0.344 | 0.294 | 0.298 | 0.217 | 0.299 | 0.218 | 0.304 | 0.225 | 0.279 | 0.183 | 0.285 | 0.190 | **0.226** | **0.128** | *0.160* | *0.062* | *0.109* | *0.028* | *0.105* | *0.027* |
| ETTm2 | 12.5% | 0.235 | 0.123 | 0.218 | 0.107 | 0.218 | 0.108 | 0.172 | 0.074 | 0.168 | 0.072 | 0.187 | 0.084 | 0.146 | 0.059 | 0.152 | 0.067 | **0.122** | **0.037** | *0.097* | *0.027* | *0.080* | *0.018* | *0.076* | *0.017* |
| | 25.0% | 0.260 | 0.138 | 0.224 | 0.113 | 0.223 | 0.113 | 0.180 | 0.081 | 0.175 | 0.077 | 0.193 | 0.090 | 0.151 | 0.068 | 0.157 | 0.072 | **0.127** | **0.040** | *0.106* | *0.031* | *0.085* | *0.020* | *0.080* | *0.020* |
| | 37.5% | 0.275 | 0.144 | 0.230 | 0.118 | 0.228 | 0.118 | 0.187 | 0.087 | 0.181 | 0.082 | 0.200 | 0.093 | 0.158 | 0.073 | 0.163 | 0.076 | **0.133** | **0.044** | *0.113* | *0.034* | *0.091* | *0.023* | *0.087* | *0.022* |
| | 50.0% | 0.291 | 0.159 | 0.235 | 0.124 | 0.233 | 0.123 | 0.196 | 0.092 | 0.190 | 0.089 | 0.205 | 0.099 | 0.163 | 0.078 | 0.167 | 0.080 | **0.138** | **0.048** | *0.120* | *0.038* | *0.098* | *0.026* | *0.095* | *0.025* |
| | Avg | 0.265 | 0.141 | 0.227 | 0.116 | 0.226 | 0.115 | 0.184 | 0.084 | 0.178 | 0.080 | 0.196 | 0.091 | 0.155 | 0.070 | 0.160 | 0.074 | **0.130** | **0.042** | *0.109* | *0.032* | *0.089* | *0.022* | *0.085* | *0.021* |
| Weather | 12.5% | 0.190 | 0.131 | 0.180 | 0.131 | 0.176 | 0.121 | 0.097 | 0.062 | 0.102 | 0.067 | 0.113 | 0.079 | 0.087 | 0.058 | 0.081 | 0.054 | **0.065** | **0.035** | *0.049* | *0.028* | *0.045* | *0.025* | *0.049* | *0.026* |
| | 25.0% | 0.195 | 0.139 | 0.190 | 0.137 | 0.180 | 0.125 | 0.107 | 0.070 | 0.108 | 0.070 | 0.128 | 0.085 | 0.093 | 0.063 | 0.084 | 0.057 | **0.068** | **0.037** | *0.055* | *0.031* | *0.052* | *0.029* | *0.052* | *0.028* |
| | 37.5% | 0.199 | 0.144 | 0.195 | 0.144 | 0.184 | 0.129 | 0.114 | 0.078 | 0.112 | 0.076 | 0.137 | 0.090 | 0.097 | 0.068 | 0.088 | 0.060 | **0.070** | **0.039** | *0.059* | *0.035* | *0.057* | *0.031* | *0.060* | *0.033* |
| | 50.0% | 0.202 | 0.151 | 0.200 | 0.151 | 0.189 | 0.133 | 0.123 | 0.086 | 0.117 | 0.077 | 0.145 | 0.095 | 0.106 | 0.070 | 0.093 | 0.064 | **0.075** | **0.043** | *0.064* | *0.038* | *0.062* | *0.034* | *0.065* | *0.037* |
| | Avg | 0.196 | 0.141 | 0.191 | 0.141 | 0.182 | 0.127 | 0.110 | 0.074 | 0.110 | 0.072 | 0.130 | 0.087 | 0.096 | 0.065 | 0.087 | 0.059 | **0.069** | **0.038** | *0.057* | *0.033* | *0.054* | *0.030* | *0.057* | *0.031* |
| Electricity | 12.5% | 0.278 | 0.146 | 0.278 | 0.146 | 0.278 | 0.146 | 0.251 | 0.127 | 0.249 | 0.124 | 0.264 | 0.132 | 0.235 | 0.112 | 0.241 | 0.118 | **0.229** | **0.100** | *0.204* | *0.083* | *0.202* | *0.085* | *0.194* | *0.080* |
| | 25.0% | 0.302 | 0.169 | 0.301 | 0.169 | 0.302 | 0.170 | 0.278 | 0.143 | 0.281 | 0.144 | 0.273 | 0.141 | 0.264 | 0.137 | 0.268 | 0.139 | **0.258** | **0.125** | *0.232* | *0.104* | *0.206* | *0.089* | *0.203* | *0.087* |
| | 37.5% | 0.324 | 0.193 | 0.323 | 0.193 | 0.325 | 0.194 | 0.295 | 0.162 | 0.319 | 0.174 | 0.294 | 0.163 | 0.287 | 0.156 | 0.293 | 0.162 | **0.283** | **0.152** | *0.258* | *0.126* | *0.213* | *0.094* | *0.211* | *0.094* |
| | 50.0% | 0.347 | 0.221 | 0.346 | 0.221 | 0.348 | 0.222 | 0.315 | 0.188 | 0.343 | 0.205 | 0.318 | 0.188 | 0.317 | 0.189 | 0.322 | 0.194 | **0.312** | **0.185** | *0.285* | *0.152* | *0.221* | *0.100* | *0.220* | *0.101* |
| | Avg | 0.313 | 0.182 | 0.312 | 0.182 | 0.313 | 0.183 | 0.285 | 0.155 | 0.298 | 0.161 | 0.287 | 0.156 | 0.276 | 0.149 | 0.281 | 0.153 | **0.270** | **0.140** | *0.245* | *0.116* | *0.211* | *0.092* | *0.207* | *0.091* |

*Table 12.* Full Accuracy Performance Comparison in Classification (red: best, blue: second, *italics*: upper bound)

| Datasets | Random | K-Center | Herding | DC | MTT | DM | CondTSC | TimeDC | UniTSC | Full | TimesNet | GPT4TS |
|---|---|---|---|---|---|---|---|---|---|---|---|---|
| EthanolConcentration | 20.2 | 18.6 | 21.7 | 23.2 | 25.1 | 26.4 | 26.2 | 26.6 | **28.9** | *31.2* | *35.7* | *34.2* |
| FaceDetection | 49.2 | 49.3 | 49.6 | 33.6 | 27.9 | 38.2 | 51.8 | 52.5 | **53.2** | *65.6* | *68.6* | *69.2* |
| Handwriting | 15.8 | 15.5 | 16.9 | 19.1 | 18.4 | 19.4 | 22.3 | 21.8 | **25.8** | *31.6* | *32.1* | *32.7* |
| Heartbeat | 49.3 | 44.9 | 50.2 | 59.5 | 59.0 | 51.6 | 63.6 | 71.2 | **73.2** | *75.6* | *78.0* | *77.2* |
| JapaneseVowels | 45.9 | 47.0 | 43.5 | 52.7 | 53.5 | 47.6 | 54.1 | 52.7 | **57.3** | *94.9* | *98.4* | *98.6* |
| PEMS-SF | 41.6 | 20.2 | 38.7 | 49.7 | 42.2 | 45.1 | 52.1 | 52.6 | **57.5** | *86.0* | *89.6* | *87.9* |
| SelfRegulationSCP1 | 70.6 | 72.7 | 64.5 | 72.4 | 71.0 | 70.6 | 81.6 | 79.9 | **87.7** | *91.8* | *91.8* | *93.2* |
| SelfRegulationSCP2 | 43.3 | 50.0 | 49.4 | 51.7 | 52.2 | 50.8 | 52.4 | 51.7 | **57.8** | *58.9* | *57.2* | *59.4* |
| SpokenArabicDigits | 25.8 | 24.2 | 30.2 | 64.0 | 58.5 | 61.6 | 64.6 | **74.8** | 72.9 | *97.1* | *99.0* | *99.2* |
| UWaveGestureLibrary | 44.1 | 31.6 | 46.6 | 70.6 | 67.5 | 69.1 | 72.8 | 71.3 | **73.1** | *82.2* | *85.3* | *88.1* |
| Avg | 40.6 | 37.4 | 41.1 | 49.6 | 47.5 | 48.0 | 54.1 | 55.5 | **58.7** | *71.5* | *73.6* | *74.0* |

*Table 13.* Full Performance Comparison in Anomaly Detection (red: best, blue: second, *italics*: upper bound)

| Models | SMAP | | | PSM | | | SMD | | | SWaT | | | MSL | | |
|---|---|---|---|---|---|---|---|---|---|---|---|---|---|---|---|
| | P | R | F1 | P | R | F1 | P | R | F1 | P | R | F1 | P | R | F1 |
| Random | 84.90 | 51.43 | 64.06 | 71.76 | 92.90 | 80.97 | 82.60 | 74.25 | 78.20 | 80.91 | 77.97 | 79.41 | 82.19 | 62.49 | 71.00 |
| K-Center | 84.86 | 51.30 | 63.94 | 73.80 | 96.33 | 83.57 | 84.72 | 75.06 | 79.60 | 80.79 | 77.90 | 79.32 | 82.11 | 62.76 | 71.14 |
| Herding | 84.97 | 51.47 | 64.11 | 78.60 | 89.26 | 83.59 | 83.98 | 74.96 | 79.21 | 81.02 | 78.08 | 79.52 | 82.15 | 62.59 | 71.05 |
| DC | 84.64 | 53.28 | 65.39 | 90.14 | 88.66 | 89.39 | 83.40 | 78.21 | 80.72 | 88.30 | 75.79 | 81.57 | 85.46 | 65.44 | 74.12 |
| MTT | 84.33 | 53.41 | 65.40 | 88.37 | 88.51 | 88.44 | 82.22 | 78.30 | 80.21 | 90.21 | 74.26 | 81.46 | 84.27 | 64.85 | 73.30 |
| DM | 85.11 | 52.98 | 65.31 | 89.84 | 90.80 | 90.32 | 84.34 | 77.19 | 80.61 | 87.67 | 77.14 | 82.07 | 83.56 | 67.40 | 74.62 |
| CondTSF | 87.32 | 53.21 | 66.13 | 93.27 | 95.43 | 94.34 | 85.70 | 78.24 | 81.80 | 90.29 | 81.36 | 85.59 | 88.58 | 66.42 | 75.92 |
| TimeDC | 86.73 | 53.06 | 65.84 | 92.12 | 96.24 | 94.13 | 84.78 | 76.17 | 80.24 | 91.45 | 79.84 | 85.25 | 87.79 | 65.09 | 74.75 |
| **UniTSC** | **89.56** | **53.86** | **67.27** | **98.90** | 92.95 | **95.83** | **91.77** | 76.35 | **83.35** | **92.12** | 80.03 | **85.66** | **90.72** | 65.43 | **76.03** |
| *Full* | *90.94* | *55.58* | *68.95* | *98.90* | *93.97* | *96.37* | *87.72* | *82.77* | *85.17* | *92.17* | *92.14* | *92.16* | *88.28* | *72.41* | *79.56* |
| *TimesNet* | *90.14* | *56.40* | *69.39* | *98.51* | *96.20* | *97.34* | *87.91* | *81.54* | *84.61* | *90.75* | *95.40* | *93.02* | *89.54* | *75.36* | *81.84* |
| *GPT4TS* | *90.60* | *60.95* | *72.88* | *98.62* | *95.68* | *97.13* | *88.89* | *84.98* | *86.89* | *92.20* | *96.34* | *94.23* | *82.00* | *82.91* | *82.45* |

*Table 14.* Full Performance Comparison in Cross-Architecture Evaluation (red: best, blue: second, *italics*: upper bound)

| Architecture | | MLP-based | | | | CNN-based | | | | Transformer-based | | | |
|---|---|---|---|---|---|---|---|---|---|---|---|---|---|
| | | MLP | | DLinear | | CNN | | TimesNet | | Transformer | | PatchTST | |
| Cond. Method | | MAE | MSE | MAE | MSE | MAE | MSE | MAE | MSE | MAE | MSE | MAE | MSE |
| ETTh1 | Random | 0.798 | 1.113 | 0.553 | 0.683 | 0.799 | 1.112 | 0.631 | 0.899 | 0.813 | 1.176 | 0.500 | 0.573 |
| | K-Center | 0.794 | 1.106 | 0.554 | 0.686 | 0.796 | 1.110 | 0.630 | 0.897 | 0.811 | 1.171 | 0.474 | 0.521 |
| | Herding | 0.794 | 1.107 | 0.546 | 0.679 | 0.792 | 1.103 | 0.626 | 0.890 | 0.804 | 1.162 | 0.467 | 0.496 |
| | DC | 0.787 | 1.082 | 0.503 | 0.567 | 0.773 | 1.066 | 0.576 | 0.765 | 0.798 | 1.087 | 0.457 | 0.475 |
| | MTT | 0.774 | 1.055 | 0.483 | 0.524 | 0.761 | 1.042 | 0.562 | 0.720 | 0.792 | 1.034 | 0.453 | 0.456 |
| | DM | 0.789 | 1.087 | 0.524 | 0.613 | 0.779 | 1.074 | 0.594 | 0.824 | 0.799 | 1.096 | 0.462 | 0.484 |
| | CondTSF | 0.758 | 0.966 | 0.434 | 0.428 | 0.739 | 0.950 | 0.527 | 0.582 | 0.784 | 0.964 | 0.439 | 0.449 |
| | TimeDC | 0.763 | 1.019 | 0.467 | 0.493 | 0.752 | 0.994 | 0.567 | 0.728 | 0.796 | 1.070 | 0.459 | 0.477 |
| | **UniTSC** | **0.746** | **0.937** | **0.428** | **0.414** | **0.728** | **0.933** | **0.510** | **0.576** | **0.773** | **0.923** | **0.414** | **0.418** |
| | *Full* | *0.716* | *0.876* | *0.400* | *0.386* | *0.696* | *0.896* | *0.402* | *0.384* | *0.742* | *0.881* | *0.398* | *0.378* |
| Electricity | Random | 0.813 | 0.984 | 0.736 | 0.794 | 0.824 | 1.000 | 0.790 | 0.953 | 0.763 | 0.812 | 0.360 | 0.279 |
| | K-Center | 0.807 | 0.965 | 0.724 | 0.762 | 0.821 | 0.976 | 0.779 | 0.906 | 0.750 | 0.798 | 0.356 | 0.274 |
| | Herding | 0.815 | 0.988 | 0.736 | 0.792 | 0.829 | 1.019 | 0.793 | 0.957 | 0.758 | 0.804 | 0.359 | 0.279 |
| | DC | 0.673 | 0.762 | 0.537 | 0.524 | 0.612 | 0.647 | 0.462 | 0.423 | 0.576 | 0.571 | 0.352 | 0.270 |
| | MTT | 0.648 | 0.721 | 0.524 | 0.502 | 0.592 | 0.586 | 0.431 | 0.374 | 0.543 | 0.522 | 0.346 | 0.264 |
| | DM | 0.689 | 0.794 | 0.584 | 0.591 | 0.634 | 0.865 | 0.494 | 0.465 | 0.594 | 0.623 | 0.353 | 0.270 |
| | CondTSF | 0.587 | 0.520 | 0.446 | 0.362 | 0.534 | 0.459 | 0.379 | 0.284 | 0.522 | 0.440 | 0.337 | 0.257 |
| | TimeDC | 0.631 | 0.575 | 0.483 | 0.412 | 0.542 | 0.470 | 0.432 | 0.387 | 0.529 | 0.448 | 0.347 | 0.266 |
| | **UniTSC** | **0.542** | **0.468** | **0.385** | **0.290** | **0.508** | **0.433** | **0.347** | **0.258** | **0.491** | **0.427** | **0.294** | **0.215** |
| | *Full* | *0.382* | *0.295* | *0.282* | *0.197* | *0.272* | *0.168* | *0.273* | *0.180* | *0.412* | *0.317* | *0.273* | *0.180* |

