# OpenReview forum: "One Batch Is Enough: A Unified Dataset Condensation Framework for General Time Series Analysis"
_ICML.cc/2026/Conference — ICML 2026 regular_

### Official Review · Reviewer_BXmG · 2026-03-02

**Soundness:** 4
**Presentation:** 3
**Significance:** 4
**Originality:** 4
**Overall Recommendation:** 5
**Confidence:** 4

**Summary:**

The paper proposes UniTSC, a unified dataset condensation framework designed for general time-series analysis across four mainstream tasks. It introduces a multi-view hybrid encoder to extract temporal, spectral, and topological features.  Additionally, it utilizes a tri-space alignment paradigm to align student and expert models across parameter, frequency, and topology spaces.  Extensive experiments on 22 datasets across four tasks demonstrate strong performance retention with extremely small synthetic sets, as well as robust cross-architecture generalization.

**Compliance With Llm Reviewing Policy:**

Affirmed.

**Final Justification:**

I keep my original score.

**Key Questions For Authors:**

See weaknesses.

**Limitations:**

Yes

**Strengths And Weaknesses:**

S1. This paper generalizes dataset condensation to four distinct time-series tasks and addresses the task bias inherent in existing time-series condensation methods. Furthermore, the extension to imputation and anomaly detection tasks broadens the scope of dataset condensation for time series.
S2. The authors provide a reasonable motivation for why parameter matching alone is insufficient and support this with theoretical analysis. In addition, extensive ablation studies, the analysis of optimization dynamics, and visualization experiments collectively demonstrate the effectiveness of the proposed components.
S3. The experimental results of UniTSC are impressive, especially its 97.9% performance retention with only 0.01% of the training data in anomaly detection tasks. Moreover, the cross-architecture evaluation convincingly demonstrates strong generalization capabilities.
S4. The framework demonstrates remarkable data efficiency and practical utility. The resulting reductions in computational and storage costs are valuable for large-scale, data-centric machine learning workflows.

W1. For the imputation and anomaly detection tasks, the paper mainly compares the proposed method with general condensation methods and coreset-based methods. Although the authors note that no task-specific condensation methods currently exist for these tasks, it would further support the claim of task bias if methods such as CondTSF could be extended and included as additional baselines for comparison.
W2. The paper uses one batch (<128 samples) as a central narrative framing, but in practice the batch budget depends on the task’s input format (e.g., sequence length). It would be better to restate this as an empirical observation under the paper’s experimental setup, rather than as a general rule.
W3. In related work A.1., the authors cited forecasting works like Nie et al., 2023; Liu et al., 2024a; Wang et al., 2024 for task-specific architectures, while works for other tasks like ReCTSi for imputation should be included.

M1. In Lines 1073–1074, the correct reference should be Fig. 5 rather than Fig. 7.

---

> ### Author Rebuttal · Authors · 2026-03-31
>
> We sincerely thank you for the highly constructive comments. Our detailed responses are provided below:
>
> ```
> W1: Extend forecasting-specific baselines (like CondTSF) for imputation to further prove the task bias claim.
> ```
>
> We appreciate this suggestion. As noted in our introduction, both CondTSF and TimeDC baselines suffer from intrinsic task biases that severely limit their imputation capabilities:
>
> - **CondTSF:** By minimizing forecasting errors, it implicitly acts as a low-pass filter. While this preserves global trends, it aggressively suppresses the high-frequency local variations and short-term volatility essential for reconstructing missing values.
> - **TimeDC:** It relies heavily on task-specific supervision signals to guide its condensation process. This intrinsic coupling with downstream objectives prevents the comprehensive preservation of underlying generative mechanisms necessary for imputation.
>
> To empirically demonstrate this, **we followed the reviewer's suggestion and extended CondTSF and TimeDC to the imputation task**. Specifically, we adapted CondTSF and TimeDC for the imputation task by leveraging a masking mechanism and a reconstruction head. The MSE results under a 25% mask ratio are summarized below:
>
> |         | ETTh1 | ETTh2 | ETTm1 | ETTm2 | Weather | Electricity |
> | :-----: | :---: | :---: | :---: | :---: | :-----: | :---------: |
> | CondTSF | 0.237 | 0.134 | 0.170 | 0.068 |  0.063  |    0.137    |
> | TimeDC  | 0.241 | 0.139 | 0.169 | 0.072 |  0.057  |    0.139    |
> | UniTSC  | 0.198 | 0.078 | 0.128 | 0.040 |  0.037  |    0.125    |
>
> As expected, **CondTSF yields sub-optimal imputation results** because its low-pass filtering nature sacrifices local structural fidelity, while TimeDC  **fails to preserve the underlying generative mechanisms** due to its heavy reliance on task-specific supervision. We will include both CondTSF and TimeDC as baselines for both imputation and anomaly detection in the final manuscript to ensure a fairer comparison and reinforce our core motivation regarding task bias.
>
> ```
> W2: Restate "one batch" as an empirical observation tied to sequence length setups rather than a general rule.
> ```
>
> We sincerely appreciate the valuable feedback. The specific budget required for effective condensation is not a universal constant, but is inherently coupled with the task's specific input format, such as the look-back sequence length (e.g., 96 or 100) or class distribution constraints (e.g., 5-shot).
>
> Our intention with the "one batch (< 128 samples)" narrative was to emphasize the extreme data efficiency observed across the 22 datasets evaluated in our studies. However, we acknowledge that presenting this as a general rule overstates the theoretical bounds of the method and could be misleading for tasks with different scale requirements.
>
> To prevent misinterpretation, we will tone down the generalized phrasing in the Introduction, explicitly restating "one batch" as an empirical observation tied to our experimental setups. Furthermore, we will also add a discussion in the **Limitations** section acknowledging that for tasks with significantly longer input sequences or highly complex distributions, the optimal batch budget need to scale proportionally with the task's input format.
>
> ```
> W3: Broaden related work citations for non-forecasting tasks (e.g., ReCTSi for imputation).
> ```
>
> Thanks for the suggestion. In the revised manuscript, **we will expand the scope of the Related Work** section. Specifically, we will properly introduce ReCTSi [Lai et al., 2024] and other methods for imputation, classification, and anomaly detection. This expansion will provide a much more balanced and comprehensive overview of the general time-series analysis landscape, better contextualizing the universal motivation behind UniTSC.
>
> ```
> M1: Fix typo in Lines 1073–1074 (reference should be Fig. 5, not Fig. 7).
> ```
>
> We sincerely thank the reviewer for pointing out this error. We confirm that it is a typographical mistake and will carefully revise the manuscript to improve its overall clarity and presentation.

---

> > ### Author Rebuttal · Reviewer_BXmG · 2026-04-02
> >
> > Fully resolved - My concerns have been adequately addressed.

---

### Official Review · Reviewer_2Uau · 2026-03-09

**Soundness:** 3
**Presentation:** 2
**Significance:** 3
**Originality:** 4
**Overall Recommendation:** 4
**Confidence:** 3

**Summary:**

This paper introduces UniTSC, a unified dataset condensation framework designed to overcome the "task bias" and feature mismatch prevalent in existing time-series condensation methods. Instead of optimizing for a specific downstream objective, UniTSC aims to preserve the invariant physical generative mechanisms of the original data. It achieves this through a multi-view hybrid encoder that extracts temporal, spectral, and topological representations, coupled with a novel tri-space alignment paradigm that jointly matches parameter trajectories, frequency distributions, and multivariate topologies between expert and student models. Extensive experiments demonstrate that UniTSC successfully condenses massive datasets into a single batch equivalent while retaining highly competitive performance across diverse tasks, including forecasting, imputation, classification, and anomaly detection.

**Compliance With Llm Reviewing Policy:**

Affirmed.

**Final Justification:**

The quality of the paper is solid, and the rebuttal has addressed my concerns.  I would like to maintain my score as Weak Accept.

**Key Questions For Authors:**

* Regarding the multi-scale patching mechanism, while the motivation is intuitive, the patch sizes are simply fixed at 16 and 8. It's questionable if such a simplistic and rigid setting can effectively adapt to time series data with varying sampling frequencies. Could the authors explain why these specific values were chosen and how sensitive the framework's overall condensation performance is to different patch sizes across diverse datasets?

* There is a discrepancy between Section 4.1 and Figure 2 regarding $X_{syn}$. The text states that a sliding window is applied over $X_{syn}$ to construct the synthetic dataset, but Figure 2 shows the sliding window being applied to the "Full Dataset" to generate $X_{syn}$. Could the authors clarify the exact initialization and generation process of $X_{syn}$?

* Figure 2 visually implies that the Spectral View takes the raw, unpatched time series as its initial input. However, the text explicitly states that the Real Fast Fourier Transform (rFFT) is applied along the patch dimension of the patch embeddings $E$. This contradiction makes it difficult to understand the exact data flow.

*  Section 4.2 states that the inverse FFT is fused via a residual connection: $\mathbf{H}' = \mathbf{H} + \text{Dropout}(\text{irFFT}(\tilde{\mathcal{F}}))$. However, the Temporal View employs multi-scale patching, yielding two distinct representations ($H^{(1)}$ and $H^{(2)}$) which are subsequently concatenated into $Z_{time}$. Figure 2 depicts only a single addition ($\oplus$) operation applied after the Transformer Encoder. Could the authors clarify exactly how and where this residual connection is applied? Specifically, is the spectral modulation added separately to both scale-specific outputs ($H^{(1)}$ and $H^{(2)}$), or is it added to the concatenated sequence $Z_{time}$?

**Limitations:**

No. Given the architectural complexity of the proposed framework, the authors should explicitly discuss the computational overhead and time efficiency of the dataset condensation process itself as a limitation. A direct comparison regarding the condensation time costs against other time-series condensation baselines should be included.

**Strengths And Weaknesses:**

## Strength

* The overall architecture and logical flow of the paper are well-structured, making the high-level concepts and the core framework easy to grasp.

* The motivation is compelling and persuasive, as addressing the "task bias" in existing dataset condensation methods is a critical and practical step forward for general time-series analysis.

* The proposed UniTSC framework is methodologically novel and comprehensive, utilizing a clever tri-space alignment to elegantly capture the underlying temporal, spectral, and topological generative mechanisms of the data.

* The experimental evaluation is extensive and solid, benchmarking the framework across four mainstream tasks and 22 datasets to convincingly demonstrate its state-of-the-art performance and cross-architecture generalizability.


## Weakness

* The presentation of the proposed methodology could be significantly improved. Specifically, the explanation of the Topological Relational View lacks necessary detail. Additionally, inconsistencies between the textual descriptions, mathematical notations, and the corresponding figures create confusion. These ambiguities make it difficult to fully grasp the exact technical derivations and data flow (detailed examples are provided in the Questions section).

* The manuscript does not explicitly detail how to determine the hyperparameters, especially the patch sizes for multi-scale modeling.

---

> ### Author Rebuttal · Authors · 2026-03-31
>
> We sincerely thank you for the valuable feedback, and our detailed responses are below:
>
> ```
> W1: The text lacks detail of Topological Relational View.
> ```
>
> Thanks for pointing this out. The Topological Relational View is designed to explicitly capture the cross-variable dependencies and multivariate coupling structures inherent in the underlying physical system. It aggregates temporal embeddings to obtain variable-level summaries and reconstructs an implicit graph structure via a self-attention mechanism. The relational features are then pooled to produce a global state vector, which is projected into a discrete topology space.
>
> Crucially, this projection acts as an information bottleneck, encouraging the framework, which can be adapted to condense datasets for different specific downstream tasks, to match discrete interaction regimes rather than stochastic fluctuations.
>
> In the final version, we will expand this section to provide explicit mathematical derivations and ensure complete consistency across the textual descriptions, notations, and architectural figures.
>
> ```
> W2&Q1: Clarify the rationale for fixing patch sizes at 16 and 8, and provide an evaluation of the model's sensitivity to different patch sizes.
> ```
>
> We sincerely thank the reviewer for raising this point. Setting the coarse patch length to 16 and the fine-grained patch length to 8 was motivated by established optimal defaults in state-of-the-art patching models (e.g., PatchTST [Nie et al., 2023]).
>
> To evaluate the sensitivity of this parameter setting, **we have conducted new forecasting experiments varying the coarse and fine patch size combinations**. The MSE results are below:
>
> |Patch Size| ETTh1 | ETTh2 | ETTm1 | ETTm2 | Weather |
> |:-:|:-:|:-:|:-:|:-:|:-:|
> |{8, 4}| 0.416 | 0.315 | 0.384 | 0.206 |  0.205  |
> |{16, 8}| 0.413 | 0.320 | 0.381 | 0.200 |  0.202  |
> |{32, 16}| 0.417 | 0.314 | 0.382 | 0.203 |  0.206  |
>
> The results show that a small patch size causes the representation to degenerate into point-level inputs, failing to capture local fluctuation semantics. Conversely, an overly large patch size encapsulates too much complex information, over-smoothing high-frequency features. The {16, 8} configuration strikes an optimal balance, achieving superior MSE performance across the majority of datasets.
>
> ```
> Q2: Inconsistency between Section 4.1 and Figure 2 regarding X_syn initialization.
> ```
>
> Thanks for pointing this out. We would like to clarify that the discrepancy arises because the two descriptions actually refer to ​**different stages**:
>
> - **Figure 2 (initialization)**: To incorporate task-adaptive priors during Synthetic Initialization, a sliding window is applied to the Full Dataset to initialize the continuous latent tensor $\mathcal{X}\_{syn}$. Here, $\mathcal{X}\_{syn}$ starts as a learnable parameter derived from real data.
> - **Section 4.1 (condensation loop)**: During optimization, a sliding-window operator is dynamically applied over the learnable tensor $\mathcal{X}\_{syn}$ to construct the synthetic dataset $\mathcal{S}$ for subsequent gradient and distribution matching.
>
> Thus, both are accurate, but they describe different stages of the process. To avoid potential confusion, we will revise Section 4.1 and add explicit annotations to Figure 2 to clearly distinguish between these two stages.
>
> ```
> Q3: Clarify the mismatch between the text and Figure 2 regarding the Spectral View.
> ```
>
> We thank the reviewer for catching this inconsistency. The text accurately states that the rFFT is applied to the patch embeddings $\textbf{E}$ along the patch dimension, not to the raw time series. We will revise Figure 2 to explicitly depict the Spectral View branching after the patch embedding layer, ensuring alignment with our methodological description.
>
> ```
> Q4: How does the residual connection H' = H + Dropout(irFFT(\tilde{F})) apply when there are two scales H^(1) and H^(2)?
> ```
>
> Thank you for this precise question. The residual connection is applied **independently to each scale**:
>
> $\mathbf{H}^{'(1)} = \mathbf{H}^{(1)} + \text{Dropout}(\text{irFFT}(\tilde{\mathcal{F}}^{(1)}))$,
>
> $\mathbf{H}^{'(2)} = \mathbf{H}^{(2)} + \text{Dropout}(\text{irFFT}(\tilde{\mathcal{F}}^{(2)}))$.
>
> These two representations $\mathbf{H}^{'(1)}$ and $\mathbf{H}^{'(2)}$ are then concatenated to form the temporal embedding $Z_{time}$. Figure 2 currently shows a single addition operation, which is a simplification that obscures this parallel structure. We will revise Figure 2 to explicitly show both residual connections and update the notation in Section 4.2 to ensure consistency.
>
> ```
> Limitation: Explicitly discuss the computational overhead.
> ```
>
> We agree that the architectural complexity of UniTSC introduces computational overhead. As suggested, we will add a dedicated **Limitations** section to the main text explicitly discussing this overhead, incorporating the time-cost analysis from Appendix D.3  to ensure complete efficiency analysis.

---

> > ### Author Rebuttal · Reviewer_2Uau · 2026-04-01
> >
> > The rebuttal has successfully addressed my concerns. Please revised and update the manuscripts as discussed.

---

### Official Review · Reviewer_aeD5 · 2026-03-12

**Soundness:** 3
**Presentation:** 3
**Significance:** 2
**Originality:** 3
**Overall Recommendation:** 4
**Confidence:** 2

**Summary:**

This paper proposes UniTSC, a unified dataset condensation framework for general time-series analysis. To make dataset condensation applicable across diverse time-series tasks, the method consists of three main components. First, task-adaptive synthetic initialization introduces only a weak task prior. Second, a multi-view hybrid encoder is used to preserve temporal, spectral, and topological views of time series. Third, the paper proposes a tri-space alignment loss. Based on extensive experiments, the authors show that UniTSC can maintain strong downstream performance even when using an extremely small condensed training set at the scale of a single batch.

**Compliance With Llm Reviewing Policy:**

Affirmed.

**Final Justification:**

The authors have acknowledged that their strongest claim, “One Batch Is Enough”—which is even included in the title—is an overclaim in the context of classification tasks. Although they addressed all points in their rebuttal, I am maintaining my score because this is an overclaim regarding the most critical aspect.

**Key Questions For Authors:**

1. In classification, since there remains a large gap to the full-data upper bound, would it be an overclaim to say that **one batch is enough** for this downstream task as well?

2. In chapter 5 (experiment part), CondTSF and TimeDC are included as baselines for forecasting and classification, but not for anomaly detection or imputation. Is this because these methods are not applicable to those two tasks?

**Limitations:**

yes

**Strengths And Weaknesses:**

**Strengths**:

1. The paper evaluates the method on a wide range of tasks and shows that it maintains strong performance even under extreme data compression.
2. This is an interesting direction because dataset condensation for time series is still much less explored than in computer vision.
3. Beyond the main results, the paper includes many additional experiments, and the ablation studies help justify the necessity of the proposed components that are designed to reflect important properties of time series. In particular, the appendix is thorough and addresses many questions that arose while reading the main paper.


**Weaknesses**:
1. Compared to vision, the practical need for dataset condensation in time series may be less compelling in some settings, since many benchmark datasets are relatively moderate in size.
2. For classification, the gap to the full-data upper bound remains substantial, which makes the paper’s **one batch is enough** message feel somewhat less convincing for this task.
3. In the efficiency study, although the condensed dataset is much smaller, the gains in training time and storage should be interpreted carefully: the absolute benefits may depend strongly on the dataset and application scenario.

---

> ### Author Rebuttal · Authors · 2026-03-31
>
> We sincerely thank the reviewer for the valuable feedback, and our detailed responses are below:
> ```
> W1: Practical need for time series condensation may be less compelling than in vision.
> ```
> We agree that many academic time-series benchmarks are relatively moderate in size. However, real-world deployments, such as manufacturing and healthcare monitoring, often rely on millions of high-frequency sensors that continuously generate large volumes of data. In such scenarios, **approximately 33% of stored data is redundant or obsolete** [Veritas Dark Data Assessment, 2016], making large-scale time-series datasets highly inefficient.
>
> Training models on such bloated data incurs substantial computational and storage costs, **which has motivated many time-series dataset condensation studies [CondTSF (Ding et al., 2024), CondTSC (Liu et al., 2024), TimeDC (Miao et al., 2025)]** to remove redundancy while preserving essential information. Furthermore, **condensed datasets enable significant computational savings for repetitive workflows** such as architecture search and hyperparameter tuning [Dataset Distillation (Wang et al., 2018), DC (Zhao et al., 2020)]. Our **Response to Reviewer 2So4.W2\&Q2** further illustrates this.
> ```
> W2&Q1: "One batch is enough" is an overclaim for classification due to the large gap to the full-data upper bounds.
> ```
> We appreciate this constructive feedback and agree that the "One Batch Is Enough" claim is overly strong when applied to classification tasks.
>
> While our experimental results demonstrate that a single batch effectively captures the core generative mechanisms for sequence-to-sequence tasks (i.e., **forecasting, imputation, and anomaly detection**), **classification** inherently relies on defining precise decision boundaries. As analyzed in Appendix D.4, an extreme budget of 5 prototypes per class struggles to fully cover the diverse data manifolds of complex datasets (e.g., PEMS-SF) with high intra-class variance.
>
> Nevertheless, **while a gap to full-data performance remains, UniTSC still outperforms all baselines by up to 21.3% (Table 12)**, establishing it as the most effective condensation method available for this task. We view closing this performance gap in classification task as an important open problem.
>
> As suggested, we will revise the manuscript to tone down the "One batch" claim and explicitly discuss this boundary-coverage limitation in the Limitations and Conclusion section.
> ```
> W3: Absolute efficiency gains depend heavily on the raw dataset size and specific application scenarios.
> ```
> Thanks for this insightful observation. For moderate-sized academic benchmarks like the ETT subsets, the absolute savings in wall-clock training time and storage are naturally limited.
>
> However, **the practical value of these relative gains scales exponentially in real-world applications**. When processing massive industrial datasets or deploying models in resource-constrained edge environments (e.g., IoT sensors, embedded MCUs), absolute reductions in computational overhead and memory storage become mission-critical. We will add a dedicated discussion to clarify how the absolute benefits of our method scale with the original dataset size and specific deployment scenario.
> ```
> Q2: Clarify why CondTSF and TimeDC are excluded as baselines for anomaly detection and imputation.
> ```
> Thanks for the careful observation. **These baselines were initially excluded because they are intrinsically coupled to their specific downstream tasks** and are not natively applicable to imputation or anomaly detection out-of-the-box:
> - **CondTSF** is explicitly driven by a forecasting objective to minimize prediction error.
> - **TimeDC** relies on curriculum-based trajectory matching guided by specific forecasting or classification supervision signals.
>
> Overall, neither method is designed to handle the reconstruction of randomly masked local segments (imputation task) or the modeling of normal distributions for boundary detection (anomaly detection task). However, we agree that comparing against them provides valuable insights into task bias.  Therefore, we adapted CondTSF and TimeDC for the imputation task by leveraging a masking mechanism and a reconstruction head. The MSE results under a 25% mask ratio are summarized below:
>
> ||ETTh1|ETTh2|ETTm1|ETTm2|Weather|Electricity|
> |:-:|:-:|:-:|:-:|:-:|:-:|:-:|
> |CondTSF|0.237|0.134|0.170|0.068|0.063|0.137|
> |TimeDC|0.241|0.139|0.169|0.072|0.057|0.139|
> |UniTSC|0.198|0.078|0.128|0.040|0.037|0.125|
>
> As expected, **CondTSF yields sub-optimal imputation results** because its low-pass filtering nature sacrifices local structural fidelity, while TimeDC  **fails to preserve the underlying generative mechanisms** due to its heavy reliance on task-specific supervision. We will include both CondTSF and TimeDC as baselines for both imputation and anomaly detection in the final manuscript to ensure a fairer comparison and reinforce our core motivation regarding task bias.

---

> > ### Author Rebuttal · Reviewer_aeD5 · 2026-04-02
> >
> > Thank you for the detailed and helpful rebuttal. I will maintain my score.

---

### Official Review · Reviewer_2So4 · 2026-03-13

**Soundness:** 3
**Presentation:** 4
**Significance:** 4
**Originality:** 4
**Overall Recommendation:** 5
**Confidence:** 3

**Summary:**

The paper studies dataset condensation for general time-series analysis, aiming to synthesize a very small dataset that can replace the original training set across forecasting, imputation, classification, and anomaly detection.
It proposes UniTSC, with three main parts:
- task-adaptive synthetic initialization (continuous latent tensor for sequence tasks; class prototypes for classification),
- a multi-view hybrid encoder over temporal, spectral, and topological views,
- a tri-space alignment objective matching parameter, frequency, and topology spaces.

**Compliance With Llm Reviewing Policy:**

Affirmed.

**Final Justification:**

Thanks for addressing my concerns in detail. They are cleared to some extend, I retain my support for this work.

**Key Questions For Authors:**

- How much does each space contribute independently across tasks? The forecasting ablation is useful, but task-wise ablations for classification/anomaly detection would help.
- Can you clarify the practical wall-clock cost of condensation itself versus downstream savings?

**Limitations:**

No, the authors do not include limitations of their work.

**Strengths And Weaknesses:**

### Strengths
- Important and timely problem: Unified condensation for time series is meaningful in my opinion; existing time-series condensation methods are mostly task-specific, and the paper explains this clearly in the introduction and related work.
- Strong empirical scope: The experiment cover 22 datasets across 4 tasks, which is broad.
- Good supporting analyses:
   - Ablations on removing patching, multi-scale, frequency, topology, and both frequency/topology show meaningful drops.
   - Cross-architecture transfer is a real plus; Figure 4 and Table 14 suggest the condensed data is not tied to a single proxy model.

### Weaknesses
- Theoretical results are not very strong. Proposition 4.1 and Theorem 4.2 rely on strong assumptions of NTK-style dynamics, Lipschitz / smoothness of loss.
- Condensation itself is not cheap. In your appendix -- Figure 7, it shows that UniTSC is second slowest method for its condensation process. The reason might stem from tri-space matching logic.
- Some claims are too strong. “One batch is enough” is catchy, but the evidence is mixed across tasks. For example, for classification task, the gap to full-data upper bounds remains large on several datasets (Table 12).

---

> ### Author Rebuttal · Authors · 2026-03-31
>
> We sincerely thank you for the positive evaluation and constructive feedback, and our detailed responses are below:
> ```
> W1: Theoretical results rely on strong assumptions of NTK-style dynamics and Lipschitz/smoothness.
> ```
> We appreciate this concern. We acknowledge that NTK dynamics and Lipschitz/smoothness are idealizations. However, we note that these assumptions are standard theoretical tools within the dataset condensation literature (e.g., NTK in KIP [Nguyen et al., 2021], RFAD [Loo et al., 2022], and Lipschitz in DDOQ [Tan et al., 2026]), and our theoretical treatment aligns with this convention.
>
> Crucially, **Proposition 4.1 is not intended to provide a tight bound under realistic conditions**, but rather to formally motivate our tri-space design. It demonstrates why parameter-space alignment alone is intrinsically insufficient: even under idealized dynamics, the distribution gap is lower-bounded by high-frequency spectral error and topological mismatch. Theorem 4.2 then proves that minimizing $\mathcal{L}\_{freq}$ and $\mathcal{L}\_{topo}$ directly tightens the generalization upper bound.
>
> Furthermore, **these theoretical intuitions are explicitly corroborated by our ablation study (Appendix D.1)**. The severe performance degradation of the w/o Freq&Topo variant proves that explicit frequency and topology constraints are strictly necessary.
>
> In the final version, we will add a dedicated discussion in Section 4 to clarify the scope and motivating role of these assumptions.
> ```
> W2&Q2: Condensation itself is not cheap. Can you clarify the practical wall-clock cost of condensation itself versus downstream savings?
> ```
> We agree that the higher condensation cost is from tri-space alignment. However, we emphasize that condensation is a **one-time offline cost**, whereas the downstream benefits are **recurring**.
>
> To clarify the practical wall-clock trade-off, consider the Electricity dataset as an example:
>
> - **Offline Condensation Cost:** 5.08s/iter × 1000 iters = 5,080s
> - **Downstream Training (Condensed Data):** 74.35s/epoch × 50 epochs = 3,717.5s
> - **Downstream Training (Full Data):** 975.34s/epoch × 50 epochs = 48,767s
>
> As observed, a single downstream training run saves ~45,050s. In real-world scenarios requiring repeated model training, such as hyperparameter tuning or neural architecture search, these time savings compound exponentially. Furthermore, while UniTSC requires more offline processing than lightweight baselines like DM, the resulting maximization of information density and physical fidelity makes it a worthwhile trade-off. In a revision, we will highlight this.
> ```
> W3: "One Batch Is Enough" is overstated for classification given the large performance gap to full-data upper bounds.
> ```
> We thank the reviewer for this constructive feedback and agree that the "One Batch Is Enough" claim is overly strong.
>
> While our experimental results demonstrate that a single batch effectively captures the core generative mechanisms for sequence-to-sequence tasks (i.e., **forecasting, imputation, and anomaly detection**), **classification** inherently relies on defining precise decision boundaries. As analyzed in Appendix D.4, an extreme budget of 5 prototypes per class struggles to fully cover the diverse data manifolds of complex datasets (e.g., PEMS-SF) with high intra-class variance.
>
> Nevertheless, **while a gap to full-data performance remains, UniTSC still outperforms all baselines by up to 21.3% (Table 12)**, establishing it as the most effective condensation method available for this task. We view closing this performance gap in classification task as an important open problem.
>
> As suggested, we will revise the manuscript to tone down the "One batch" claim and explicitly discuss this boundary-coverage limitation in the Limitations and Conclusion section.
> ```
> Q1: Per-space ablation contributions across tasks, especially for classification and anomaly detection.
> ```
> Thanks for this excellent suggestion.  **We have added new task-wise ablation experiments for Classification and Anomaly Detection**, evaluating three variants: w/o Freq, w/o Topo, and w/o Freq&Topo.
>
> |Tasks|Dataset|w/o Freq|w/o  Topo|w/o Freq&Topo|UniTSC|
> |-|-|-|-|-|-|
> |Classification (Acc)|Heartbeat|—|69.8|—|73.2|
> ||SelfRegulationSCP1|—|81.2|—|87.7|
> |Anomaly Detection (F1)|SMAP|65.32|66.07|64.59|67.27|
> ||PSM|91.06|92.47|89.55|95.83|
>
> (Note: As stated in Section 4.3, frequency alignment is not applied to classification, hence the "—" marks ).
>
> For classification, removing the topological space severely degrades accuracy, indicating that capturing inter-variable interactions is critical for defining class separation. For anomaly detection, both frequency and topological spaces play complementary roles. Anomalies typically manifest not only as a breakdown of normal inter-variable correlations (captured topologically) but also as abrupt high-frequency shifts (captured spectrally). We will include these insights in the revised appendix.

---

> > ### Author Rebuttal · Reviewer_2So4 · 2026-04-02
> >
> > Thanks for addressing my concerns in detail. They are cleared to some extend, I retain my support for this work.

---

### Decision · Program_Chairs · 2026-04-30

**Decision:**

Accept (regular)

**Comment:**

This submission addresses an important and relatively underexplored problem: general time-series dataset condensation across forecasting, imputation, classification, and anomaly detection. On the empirical side, the paper is very strong (including many datasets and tasks) and reviewers also consistently viewed the methodological contribution as novel and well motivated, particularly the attempt to overcome task bias through a unified multi-view representation and alignment design. Several reviewers also found the practical data efficiency results compelling. After reading the initial reviews and the rebuttal, the latter appears to have resolved most of the concrete concerns.

From my perspective, the main caveat is that some of the paper’s (initial) framing is too strong relative to the evidence. In particular, the title claim “One Batch ....” is not uniformly supported across tasks, and the authors acknowledge that this is an overclaim to some extent. The theory is also more motivational than decisive, relying on standard but idealized assumptions, and the condensation process itself is not (computationally) cheap (which is somewhat of a minor issue in my opinion). Still, these concerns seem more about calibration and scope than about core validity. Overall, the rebuttal appropriately toned down the strongest claims while adding useful missing evidence. Primarily due to the "overclaiming" issue, I am recommending "Weak Accept".